# Transcriptional profiling reveals extraordinary diversity among skeletal muscle tissues

Erin E Terry[1], Xiping Zhang[2], Christy Hoffmann[1], Laura D Hughes[3], Scott A Lewis[1], Jiajia Li[1], Matthew J Wallace[1], Lance A Riley[2], Collin M Douglas[2], Miguel A Gutierrez-Monreal[2], Nicholas F Lahens[4], Ming C Gong[5], Francisco Andrade[5], Karyn A Esser[2], Michael E Hughes[1]*

[1]Division of Pulmonary and Critical Care Medicine, Washington University School of Medicine, St. Louis, United States; [2]Department of Physiology and Functional Genomics, University of Florida College of Medicine, Gainesville, United States; [3]Department of Integrative, Structural and Computational Biology, The Scripps Research Institute, La Jolla, United States; [4]Institute for Translational Medicine and Therapeutics, Perelman School of Medicine, University of Pennsylvania, Philadelphia, United States; [5]Department of Physiology, University of Kentucky School of Medicine, Lexington, United States

**Abstract** Skeletal muscle comprises a family of diverse tissues with highly specialized functions. Many acquired diseases, including HIV and COPD, affect specific muscles while sparing others. Even monogenic muscular dystrophies selectively affect certain muscle groups. These observations suggest that factors intrinsic to muscle tissues influence their resistance to disease. Nevertheless, most studies have not addressed transcriptional diversity among skeletal muscles. Here we use RNAseq to profile mRNA expression in skeletal, smooth, and cardiac muscle tissues from mice and rats. Our data set, MuscleDB, reveals extensive transcriptional diversity, with greater than 50% of transcripts differentially expressed among skeletal muscle tissues. We detect mRNA expression of hundreds of putative myokines that may underlie the endocrine functions of skeletal muscle. We identify candidate genes that may drive tissue specialization, including *Smarca4*, *Vegfa*, and *Myostatin*. By demonstrating the intrinsic diversity of skeletal muscles, these data provide a resource for studying the mechanisms of tissue specialization.

DOI: https://doi.org/10.7554/eLife.34613.001

*For correspondence:
michael.hughes@wustl.edu

**Competing interests:** The authors declare that no competing interests exist.

## Introduction

Gene expression atlases have made enormous contributions to our understanding of genetic regulatory mechanisms. The field of functional genomics was set in motion by the completion of the Human Genome Project and the coincident development of high throughput gene expression profiling technologies. The overriding goals of this field are to understand how genes and proteins interact at a whole-genome scale and to define how these interactions change across time, space, and different disease states. The development of SymAtlas was an early, influential effort to address these questions (*Su et al., 2002*). Custom microarrays were used to systematically profile mRNA expression in dozens of tissues and cell lines from humans and mice. Besides describing tissue-specific expression patterns, these data provided essential insights into the relationship between chromosomal structure and transcriptional regulation (*Su et al., 2004*).

**eLife digest** About 40% of our weight is formed of skeletal muscles, the hundreds of muscles in our bodies that can be voluntarily controlled by our nervous system. At the moment, the research community largely sees all these muscles as a single group whose tissues are virtually interchangeable.

Yet, skeletal muscles have highly diverse origins, shapes and roles. For example, our diaphragm is a long muscle that contracts slowly and rhythmically so we can draw breaths, while tiny muscles in our eyes generate the short and precise movements of our eyeballs. Different skeletal muscles also respond in distinct ways to injuries, drugs and diseases. This suggests that these muscles may be diverse at the genetic level.

While all the cells in our body have the same genetic information, exactly which genes are turned on and off (or 'expressed') changes between types of cells. On top of this 'on or off' regulation, the level of expression of a gene – how active it is – can also differ. However, the studies that examine the differences in gene expression between tissues usually overlook skeletal muscles.

Here, Terry et al. use genetic techniques to measure how genes are expressed in over 20 types of muscle in mice and rats. The results show that the expression levels of over 50% of all the animals' genes vary between muscles. In fact, any two types of muscles express on average 13% of their genes differently from each other. The analyses yield further unexpected findings. For example, the expression levels in a muscle in the foot that helps to flex the rodents' toes are more similar to those found in eye muscles than to the ones observed in limb muscles. These conclusions indicate that skeletal muscles are a widely diverse family of tissues.

The research community will be able to use the data collected by Terry et al. to explore further the origins and the consequences of the differences between skeletal muscles. This could help researchers to understand why specific groups of muscles are more susceptible to disease, or react differently to a drug. This knowledge could also be exploited to refine approaches in tissue engineering, which aims to replace damaged muscles in the body.

DOI: https://doi.org/10.7554/eLife.34613.002

Related approaches have had a similarly high impact on biomedical research. For example, microRNA expression throughout mammalian tissues was described in an expression atlas that has revolutionized the study of regulatory RNAs (*Landgraf et al., 2007*). Our lab has contributed to this growing literature with the creation of CircaDB, a database of tissue-specific mRNA rhythms in mice (*Hughes et al., 2009*; *Zhang et al., 2014*). Taken together, these projects demonstrate that publicly available functional genomics data have enduring value as a resource for the research community.

Nevertheless, previous gene expression atlases have largely ignored skeletal muscle. CircaDB includes gene expression data from the heart and whole calf muscle, but it does not distinguish between their constituent tissues. Similarly, SymAtlas profiles nearly 100 different tissues, but there is only a single representative sample for either cardiac or skeletal muscle. The microRNA Atlas includes over 250 human, mouse, and rat tissues; however, skeletal muscle is entirely absent from these data. More recent human gene expression atlases have similar biases (*Evangelista et al., 2015*; *Lindholm et al., 2014*; *Vissing and Schjerling, 2014*). Many studies have compared muscle-specific gene expression in different tissues (*Porter et al., 2001*), fiber-types (*Chemello et al., 2011*), or disease states (*Chen et al., 2000*; *Colantuoni et al., 2001*), but on the whole, there is no systematic analysis of transcriptional diversity in skeletal muscle.

This gap in the literature is problematic since skeletal muscle comprises a remarkably diverse group of tissues (*Tirrell et al., 2012*). Skeletal muscle groups originate from different regions of the developing embryo, and they have characteristic morphological specializations (*Merrell and Kardon, 2013*; *Murphy and Kardon, 2011*; *Noden and Francis-West, 2006*). Their physiological functions are similarly diversified. For example, extraocular muscles govern precise eye movements, the diaphragm drives rhythmic breathing, and limb skeletal muscles are involved in either fast bursts of motion or sustained contractions underlying posture.

These intrinsic differences contribute to differential susceptibility of muscle groups to injury and disease. For example, there are six major classes of muscular dystrophy, and each one afflicts a

characteristic pattern of skeletal and cardiac muscle tissues (*Ciciliot et al., 2013*; *Emery, 2002*). Since the causative mutations underlying congenital muscular dystrophies are germline and present in all cells, this observation indicates that there are properties intrinsic to different muscle tissues that regulate their sensitivity or resistance to different pathological mechanisms.

Acquired diseases show similar specificity in which muscle tissues they affect and which they spare. Patients with chronic obstructive pulmonary disorder (COPD) typically have pronounced myopathy in their quadriceps and dorsiflexors (*Barreiro and Gea, 2016*; *Clark et al., 2000*; *Gagnon et al., 2013*). Critical illness myopathy, a debilitating condition caused by mechanical ventilation and steroid treatment, causes muscle weakness in limb and respiratory muscles while sparing facial muscles (*Aare et al., 2011*; *Hermans and Van den Berghe, 2015*; *Latronico et al., 2012*). Cancer cachexia (*Acharyya et al., 2005*), HIV (*Serrano et al., 2008*), and sepsis (*Tiao et al., 1997*) cause muscle wasting, typically affecting fast twitch fibers more severely than slow twitch fibers. In fact, histologically identical muscle fibers show widely divergent responses to injury and disease depending on the muscle group in which they reside (*Ciciliot et al., 2013*; *Aravamudan et al., 2006*). Taken together, these observations strongly suggest that the intrinsic diversity of skeletal muscle has important consequences for human health and disease. However, the mechanisms through which disease susceptibility and functional specialization are established are unknown.

Here we present the first systematic examination of transcriptional programming in different skeletal muscle tissues. We find that more than 50% of transcripts are differentially expressed among skeletal muscle tissues, an observation that cannot be explained by fiber type composition or developmental history alone. We show conservation of gene expression profiles across species and sexes, suggesting that these data may reveal conserved functional elements relevant to human health. Finally, we discuss how this unique data set may be applied to the study of disease, particularly regarding muscular dystrophy and regenerative medicine.

## Results

To determine which skeletal muscle tissues are of the broadest general interest, we distributed a Google poll (*Figure 1—figure supplement 1*) to leading investigators in the skeletal muscle field. Having recorded over 100 individual responses, we selected 11 mouse skeletal muscle tissues (*Table 1*) for study in order to span the functional, developmental, and anatomical diversity of skeletal muscles. To hedge against selection bias from the investigators asked to vote in the poll, we cross-correlated these results with papers indexed in NCBI's PubMed (*Figure 1—figure supplement 1*). In addition to these 11 mouse skeletal muscle tissues, we also identified representative smooth and cardiac muscle tissues for collection from mouse, and two skeletal muscle tissues (*EDL* and *soleus*) from male and female rats to permit inter-species and inter-sex comparisons (*Table 1*).

Adult mice and rats were sacrificed and whole muscle tissues were dissected to include the entire muscle body from tendon to tendon. Each sample included six biological replicates from three animals apiece. Therefore, tissues were collected from 18 individual animals for every sample. RNA was purified from these tissues, and RNAseq was used to measure global gene expression (Materials and methods and *Supplementary file 1*). On average, every muscle sample was covered by greater than 200 million aligned short nucleotide reads, for a grand total of over 4.4 billion aligned reads in the entire data set (*Table 1*). Empirical simulations indicate that this experimental design reaches saturation with respect to identification of expressed exons (*Figure 1—figure supplement 2*). Pairwise comparisons of each replicate sample further indicate a high degree of reproducibility in expression level measurements among biological replicates (median $R^2$ value >0.93, *Figure 1—figure supplement 3*).

Over 80% of transcripts encoded in the genome are expressed in at least one skeletal muscle (*Figure 1A*). Comparing the transcripts expressed in all tissues identifies a core group of ~21,000 transcripts found in every skeletal, cardiac, and smooth muscle (*Figure 1B*). Presumably, these mRNAs include the minimal set of genes required for a cell to generate contractile force. Differential expression analysis shows that even at extremely stringent ($q < 10^{-6}$) statistical thresholds, 55.2% of mouse transcripts are differentially expressed among skeletal muscle tissues (*Figure 1C*). Phrased differently, over half of all transcripts are statistically different when comparing mRNA expression among the 11 mouse skeletal muscles in this study. To validate a subset of these data, we selected 10 genes differentially expressed between *EDL* and *soleus* across a range of different fold changes

**Table 1.** The biological samples collected in this study and the total number of aligned RNAseq reads.

| Tissue | Type | Species | Sex | Replicates | Aligned reads (Millions) |
|---|---|---|---|---|---|
| Total Aorta | Smooth | Mouse | Male | 6 | 205.2 |
| Abdominal Aorta | Smooth | Mouse | Male | 6 | 219.7 |
| Thoracic Aorta | Smooth | Mouse | Male | 6 | 214.0 |
| Atria | Cardiac | Mouse | Male | 6 | 169.0 |
| Left Ventricle | Cardiac | Mouse | Male | 6 | 193.9 |
| Right Ventricle | Cardiac | Mouse | Male | 6 | 176.2 |
| Diaphragm | Skeletal | Mouse | Male | 6 | 159.1 |
| EDL | Skeletal | Mouse | Male | 6 | 176.8 |
| Extraocular | Skeletal | Mouse | Male | 6 | 171.7 |
| FDB | Skeletal | Mouse | Male | 6 | 202.5 |
| Masseter | Skeletal | Mouse | Male | 6 | 216.2 |
| Plantaris | Skeletal | Mouse | Male | 6 | 185.1 |
| Soleus | Skeletal | Mouse | Male | 6 | 172.2 |
| Tongue | Skeletal | Mouse | Male | 6 | 210.4 |
| Gastrocnemius | Skeletal | Mouse | Male | 6 | 276.1 |
| Quadriceps | Skeletal | Mouse | Male | 6 | 275.6 |
| Tibialis Anterior | Skeletal | Mouse | Male | 6 | 276.6 |
| EDL | Skeletal | Rat | Male | 6 | 226.4 |
| EDL | Skeletal | Rat | Female | 6 | 206.8 |
| Soleus | Skeletal | Rat | Male | 6 | 261.2 |
| Soleus | Skeletal | Rat | Female | 6 | 243.0 |
| Average | | | | 6 | 211.3 |
| Total | | | | 126 | 4,437.7 |

DOI: https://doi.org/10.7554/eLife.34613.003

and performed quantitative PCR (qPCR) on independent biological samples. All 10 replicated, and manual examination of internal controls for cardiac and smooth muscle agreed with *a priori* expectations (*Figure 1—figure supplement 4*). To explore similarity between tissues, we calculated a pairwise Euclidean distance between every tissue (*Figure 1—figure supplement 5*). From these data, we generated a dendrogram that clusters tissues based on overall transcriptome similarity (*Figure 1D*). Smooth and cardiac tissues clustered together as expected, and skeletal muscles were found in two primary clusters, presumably based on the proportion of fast twitch fibers they include (i.e. the proportion of *Myosin heavy chain 4* (*Myh4*) expression). We then calculated the proportion of differentially expressed transcripts in every pairwise tissue comparison (*Figure 1E*). Some surprising observations emerged from this analysis. For example, masseter, a head/neck muscle, is more similar to limb muscles like *EDL* than muscles that share a similar developmental history, such as the tongue. In contrast, the *flexor digitorm brevis* (*FDB*), a muscle necessary for flexing the toes, is more similar to extraocular muscles than limb muscles. On average, 13% of transcripts are differentially expressed between any two skeletal muscles, with a maximum of 36.5% and a minimum less than 1%. A list of all transcripts differentially expressed among skeletal muscles is provided in *Supplementary file 2*, and the analyzed expression data can be found in our online database.

Skeletal muscle contains numerous non-muscle cells types. Therefore, a key question is whether the transcriptome diversity described above reflects genuine expression differences in muscle cells or alternatively differential cellular composition of the bulk tissues. To answer this we performed principal component analysis (*Figure 2*). The first principal component (PC1) accounts for nearly 80% of the variance in our data and separates skeletal muscles into two groups with striking similarity to skeletal muscle clusters 1 and 2 in *Figure 1E*. The second principal component (PC2) accounts for roughly 8% of variance and separates cardiac and smooth muscles from skeletal muscle (*Figure 2A*).

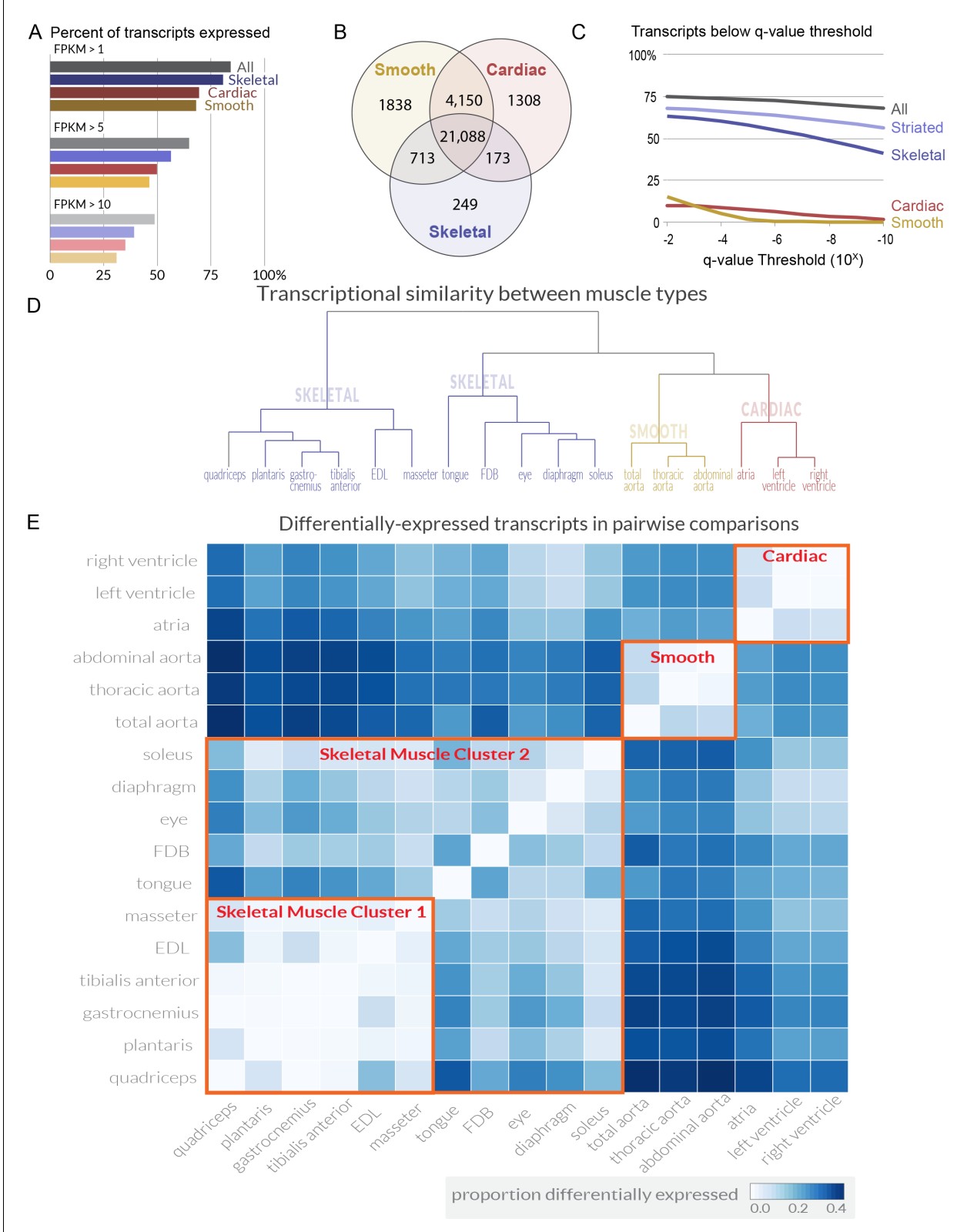

**Figure 1.** Transcriptome profiling reveals extensive gene expression differences among muscle tissues. (**A**) The percent of all transcripts detected as being expressed in different classes of tissues is shown as a bar graph. >80% of all transcripts are detectably expressed in at least one skeletal muscle tissue. (**B**) The number of transcripts expressed (FPKM >1) in *every* cardiac, smooth, or skeletal muscle tissue is shown as a Venn diagram. A core of ~21,000 transcripts is expressed in every contractile tissue. Please note that panel A describes transcripts expressed in at least one tissue; panel B

*Figure 1 continued on next page*

*Figure 1 continued*

describes transcripts expressed in every tissue. (C) The percent of transcripts showing differential expression between tissues is shown at different false-discovery rate (q-value) thresholds. One-way ANOVAs of all tissues, all striated, all skeletal, all cardiac, and all smooth muscles were used to calculate q-values. Striated muscle refers to skeletal plus cardiac muscles. (D) The overall similarity of transcriptional profiles in different tissues is displayed as a dendrogram. Notably, the three major classes of muscle (smooth, cardiac, and skeletal) cluster together as expected. (E) The number of differentially expressed transcripts (q < 0.01, fold change >2) in pairwise comparisons is shown as a heat map. Red boxes indicate clustering by similarity of (1) cardiac muscle, (2) smooth muscle, and (3) two different clusters of skeletal muscle.

DOI: https://doi.org/10.7554/eLife.34613.004

The following figure supplements are available for figure 1:

**Figure supplement 1.** Tissues were selected for transcriptional profiling to maximize the utility of these data to the skeletal muscle field.

DOI: https://doi.org/10.7554/eLife.34613.005

**Figure supplement 2.** Empirical simulations show that the read depth of this study approaches saturation for detecting expressed exons.

DOI: https://doi.org/10.7554/eLife.34613.006

**Figure supplement 3.** FPKM values show high reproducibility between replicate samples.

DOI: https://doi.org/10.7554/eLife.34613.007

**Figure supplement 4.** Internal controls demonstrate the reliability of differential expression analysis.

DOI: https://doi.org/10.7554/eLife.34613.008

**Figure supplement 5.** Euclidean distance measurements support the clustering of muscle transcriptomes into four distinct groups.

DOI: https://doi.org/10.7554/eLife.34613.009

**Figure supplement 6.** Subsets of genes are expressed specifically in different skeletal muscles.

DOI: https://doi.org/10.7554/eLife.34613.010

**Figure supplement 7.** Deep sequencing the muscle transcriptome reveals numerous novel splicing events.

DOI: https://doi.org/10.7554/eLife.34613.011

We then calculated the correlation (expressed as $R^2$ values) between each transcript's expression and PC1 (*Figure 2B*). A small number of transcripts are highly ($R^2$ >0.90) correlated with PC1 and are thereby predictive of skeletal muscle identity. From the literature, we identified a number of 'marker' genes commonly used to separate non-muscle cells from skeletal muscle in flow cytometry (*Liu et al., 2015*). We then compared the expression these 'marker' genes to PC1 to identify non-muscle cell types that may be partially responsible for the overall transcriptome diversity in our data. *Pecam1* (CD31), a marker of endothelial cells, has an $R^2$ value of 0.56, reflecting modest correlation with PC1. These data suggest that different proportions of endothelial cells in skeletal muscle contribute in part to the overall mRNA diversity observed.

All other known marker genes had lower $R^2$ values than *Pecam1*/CD31, reflecting weaker correlation with PC1. These include markers of muscle stem cells (*Vcam1*, $R^2$ = 0.37), fibroblasts (*Ly6a*/Sca1, $R^2$ = 0.19), and blood cells (*Ptprc*/CD45, $R^2$ <0.01). This observation supports the interpretation that some gene expression differences observed in this study are due to differential cellular composition. Nevertheless, every one of the top ten genes most strongly correlated with PC1 (*Figure 2D*) encodes a characteristic skeletal muscle protein that is unlikely to have been transcribed by non-muscle cell types. To extend on this observation, *Supplementary file 3* provides a list of the top 100 genes most strongly correlated with PC1. GO pathway analysis (*Huang et al., 2009a*) reveals that the most significantly enriched term among this list of genes is 'muscle protein' (enrichment 6.58, FDR < $10^{-46}$). Moreover, some of these genes are positively correlated with PC1 while others are negatively correlated. This observation implies that PC1 is more than simply a gross measure of the relative contribution of muscle cell RNA in any given sample. Taken as a whole, these observations indicate that many of the gene expression differences in this study reflect bona fide differences among muscle cell transcriptional programs rather than contamination from non-muscle mRNAs.

To extend on this analysis, we performed Gene Ontology (GO) analysis (*Huang et al., 2009a*) on every pairwise comparison of skeletal muscle tissues. We then consolidated these results by identifying GO terms that were repeatedly enriched in pairwise comparisons. *Figure 2—source data 1* summarizes these results for all 11 skeletal muscle tissues. Consistent with the principal component analysis described above, many of these GO terms were related to structural components of the sarcomere, including Z-disc, A-band, M-band, and others. We also observed enrichment in terms related to extracellular proteins such as Integrin, Collagen, and Fibronectin. More general terms were also enriched, including various pathways involved in mitochondrial function, fatty acid

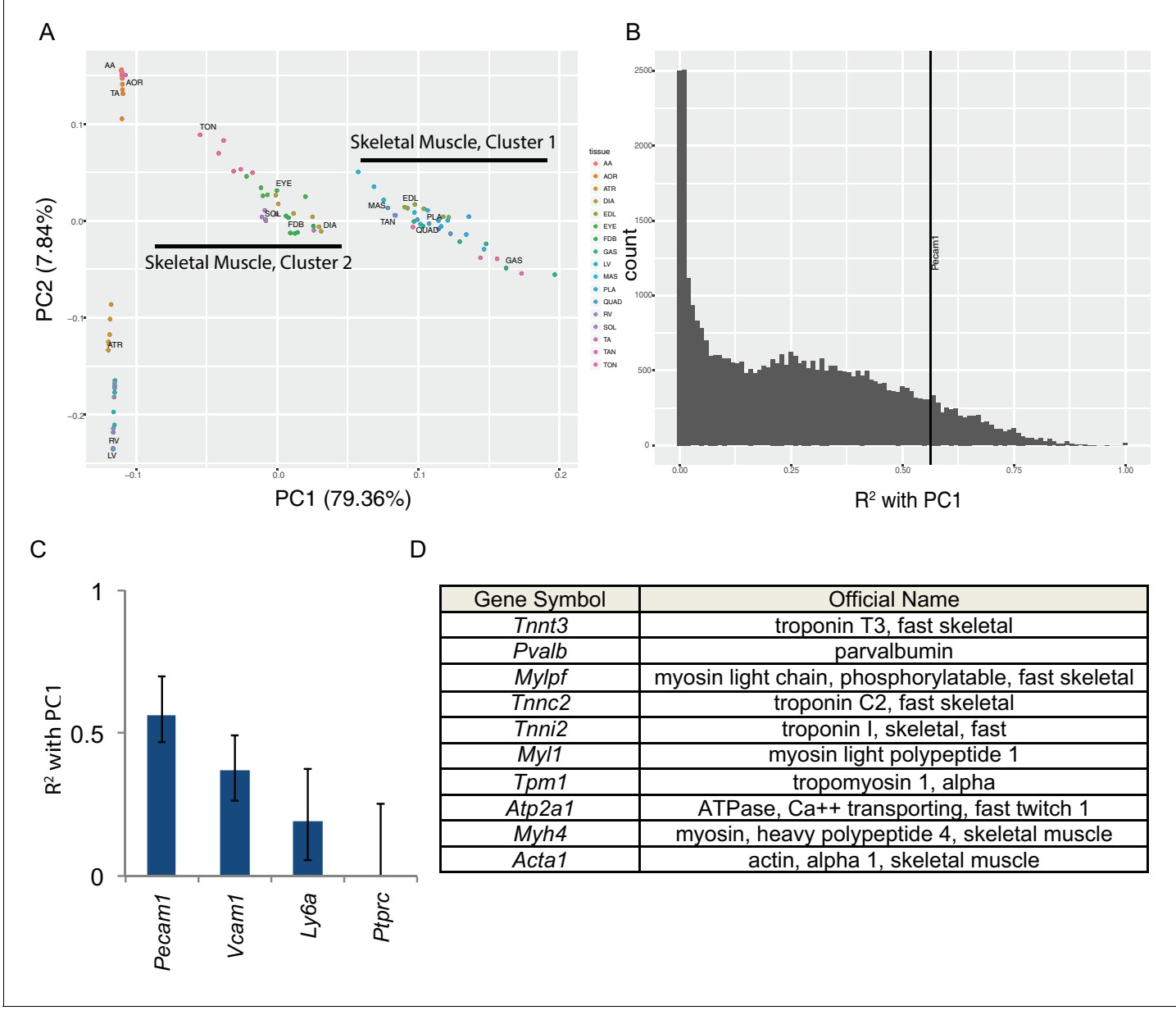

**Figure 2.** Principal component analysis reveals the majority of variance in these data is due to gene expression in skeletal muscle cells. (A) Principal components 1 and 2 are plotted on the x- and y-axes respectively for all six replicates of each muscle tissue. PC1 accounts for 79.36% of the variance and separates skeletal muscle cluster 1 from cluster 2 (compare with *Figure 1D*). PC2 accounts for 7.84% of the variance in these data and largely separates cardiac from smooth muscle tissues. (B) Histogram of the $R^2$ values of each transcript's correlation with PC1. The most correlated non-muscle cell 'marker' gene, *Pecam1*, is denoted as a vertical line. (C) $R^2$ values with 99% confidence intervals are shown for four 'marker' genes conventionally used to identify non-skeletal muscle cells in whole muscle preparations. (D) The top ten genes whose expression is most highly correlated with PC1 are shown as a Table. All ten are characteristic skeletal muscle genes.

DOI: https://doi.org/10.7554/eLife.34613.012

The following source data and figure supplement are available for figure 2:

**Source data 1.** Gene Ontology analysis reveals that sarcomere structural components are among the most commonly enriched differential pathways among skeletal muscles.
DOI: https://doi.org/10.7554/eLife.34613.014

**Figure supplement 1.** WGCNA analysis reveals co-expressed modules of genes in different muscle tissues.
DOI: https://doi.org/10.7554/eLife.34613.013

metabolism, and neuromuscular junction assembly. To extend on these observations, we performed co-expression analysis using WGCNA (*Langfelder and Horvath, 2008*). As an internal control, we first identified clusters of genes that differentiate smooth, cardiac, skeletal muscle cluster one and skeletal muscle cluster 2 (*Figure 2—figure supplement 1A*). As expected, GO analysis of the most statistically significant clusters of co-expressed genes (*Supplementaryl file 4*) included genes conventionally associated with smooth, cardiac, and skeletal muscle physiology respectively. Co-expression analysis of all 17 mouse tissues revealed a smaller number of statistically significant gene clusters (*Figure 2—figure supplement 1B*), and analysis of the resulting gene lists revealed enrichment in genes typically believed to be specific for skeletal muscle tissue (*Supplementary file 5*). These observations are consistent with the notion that many of the gene expression differences observed herein are genuine products of differential muscle cell gene expression. Moreover, close examination of the lists of genes identified by principal component, gene ontology, and co-expression analyses reveals that the most informative genes regarding skeletal muscle identity come from families of genes known for their function in skeletal muscle, such as *Troponin*, *Tropomyosin*, *Calsequestrin*, *Myosin heavy chain* (*Myh*), and *Myosin light chain*.

Skeletal muscle is comprised of myofibers which are typically classified into one of four types in mice based on their *Myh* expression (*Haizlip et al., 2015*). We used our data to quantify the relative abundance of different fiber types across skeletal muscle tissues (*Figure 3A*). These results were cross-correlated with legacy data from histological studies, showing close agreement (*Figure 3—figure supplement 1*). One hypothesis is that fiber type composition (i.e. the relative amounts of fast versus slow twitch fibers) establishes the observed diversity of gene expression profiles. To test this, we clustered skeletal muscle tissues based on similarity of *Myh* expression. The resulting dendrograms are grossly similar to those based on global gene expression (*Figure 3B*; compare with *Figure 1D*), as predominantly fast twitch muscles expressing high levels of *Myh4* (Type IIB fibers) tend to cluster together. Nevertheless, clustering within the two major skeletal muscle groups reveals important differences, such as the high similarity of diaphragm and *FDB* based on *Myh* expression, but their dissimilarity based on global gene expression (*Figure 1E*). Similarly, the clustering of tongue and extraocular eye muscles varies considerably depending on whether *Myh* or global gene expression establishes the pairwise Euclidean distance.

To explore these observations further, we made use of a single-fiber microarray study that identified genes enriched in slow (Type I) versus fast twitch (Type IIB) skeletal muscle fibers (*Chemello et al., 2011*). These legacy data reveal that *Myostatin* (*Mstn*) is almost exclusively expressed in fast twitch fibers. Therefore, if fiber type composition determines gene expression patterns, we would expect close correlation between *Mstn* and *Myh4*. In actuality, there is only moderate correlation ($R^2$ = 0.47, *Figure 3C*). Expanding on this result, we find weak correlation between genes enriched in fast twitch fibers and *Myh4* (median $R^2$ = 0.132, *Figure 3D*). The low correlation between fiber type composition and gene expression signatures is also seen in slow twitch fibers (*Figure 3E and F*). Taken together, we conclude that fiber type based on *Myh* expression contributes to tissue-specific gene expression but is insufficient to establish the diversity of transcriptional patterns we observe.

Alternatively, developmental history may play an essential role in defining gene expression patterns in adult skeletal muscle. *Hox* genes are a family of transcription factors that establish anterior/posterior patterning and skeletal muscle specification during development (*Krumlauf, 1994*). We therefore examined the expression of *Hox* genes in muscle tissues (*Figure 4A*). Clustering of tissues based on *Hox* gene expression reveals that the head and neck muscles cluster more closely with cardiac tissues than other skeletal muscles. Similarly, the diaphragm is more similar to the aorta than limb skeletal muscle. This is in marked contrast to clustering by the entire transcriptome, where each skeletal muscle is more similar to other skeletal muscles than any cardiac or smooth tissue. *Hox* gene expression in muscle recapitulates the developmental history of these tissues with respect to anterior/posterior axis formation. But, as these clusters are significantly different from those seen when comparing whole transcriptome expression patterns (*Figure 1D*), we conclude that *Hox* genes are insufficient to explain the mRNA diversity among adult skeletal muscle tissues.

Similar to *Hox* family members, many developmentally significant genes maintain expression in adult muscle tissues. These include the myogenic regulatory factor *Myf6* that is expressed in, and exclusive to, skeletal muscle (*Figure 4B*). Related differentiation factors, *Myod1*, *Myf5*, and *Myog*, are also expressed in adult muscle, albeit at roughly 10-fold lower levels than *Myf6*. *Hotair* is a

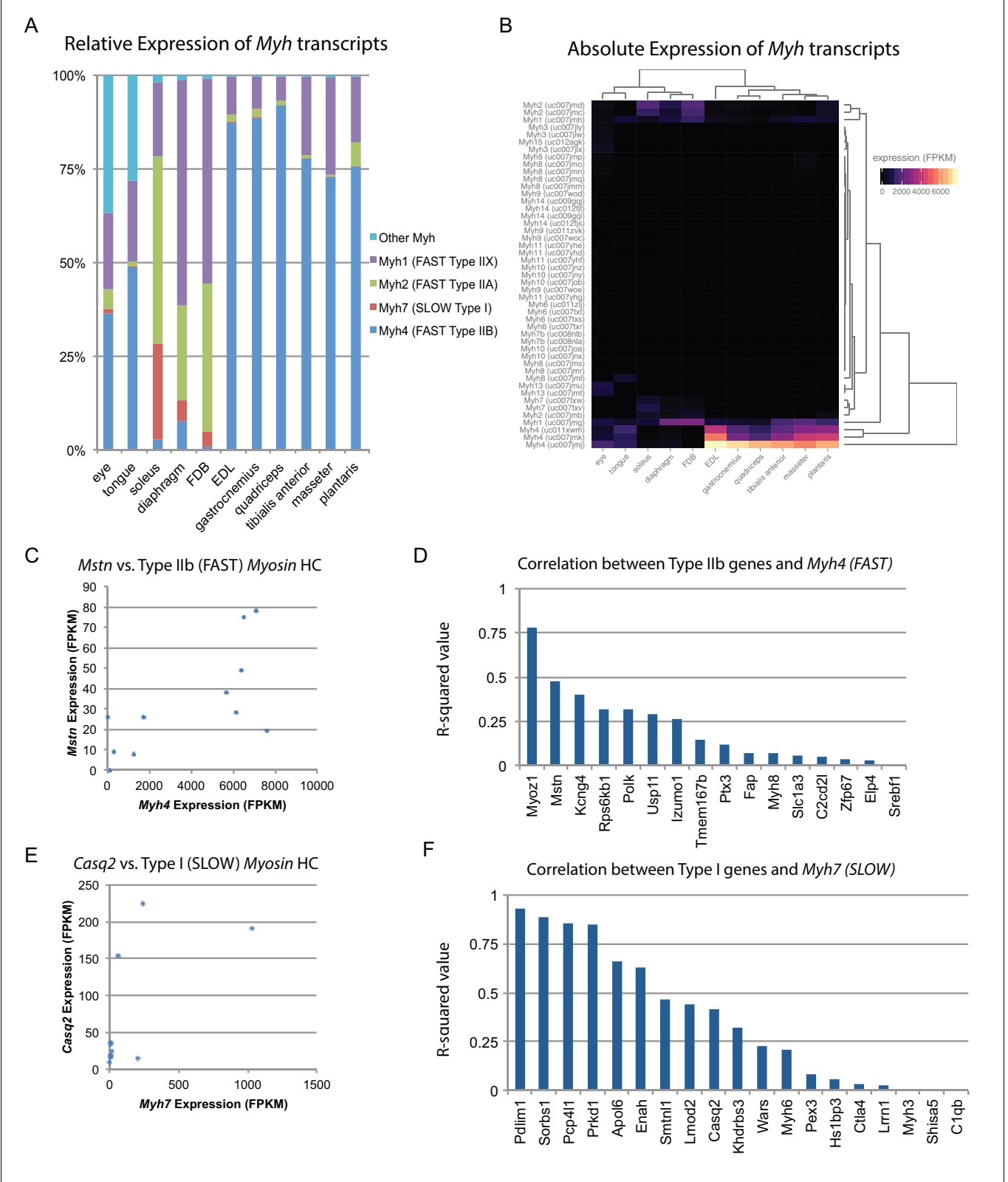

**Figure 3.** Fiber-type composition can explain some, but not all, transcriptional variability between skeletal muscle tissues. (**A**) Relative expression of *Myosin heavy chain* (*Myh*) transcripts as distributed among all 11 mouse skeletal muscle tissues is plotted as a bar graph. The y-axis describes the percent of reads aligning to any given *Myh* transcript relative to all *Myh*-aligning reads. (**B**) Absolute expression of *Myh* transcripts among all 11 mouse skeletal muscle tissues is represented as a heat map. Absolute levels were plotted as a heat map to illustrate the dynamic range of *Myh* expression in

*Figure 3 continued on next page*

*Figure 3 continued*

muscle. Tissues are clustered by overall *Myh* similarity (top dendrogram). Notably, clustering is similar to, but distinct from, clustering done on global transcriptional profiles (*Figure 1D*). (C) The expression of *Myostatin* (*Mstn*), a gene specifically expressed by Type IIb (fast) fibers (*Chemello et al., 2011*), is plotted versus *Myh4* expression. Each dot represents one of 11 skeletal muscle tissues ($R^2 = 0.473$). (D) R-squared values for correlations with *Myh4* expression are plotted as a bar graph for 16 genes known to be specific for Type IIb fibers. Median $R^2 = 0.132$; only one gene has an $R^2 > 0.5$. (E) The expression of *Calsequestrin 2* (*Casq2*), a gene specifically expressed by Type I (slow) fibers (*Chemello et al., 2011*), is plotted versus *Myh7* epression. Each dot represents one of 11 skeletal muscle tissues ($R^2 = 0.412$). (F) R-squared values for correlations with *Myh7* expression are plotted as a bar graph for 19 genes known to be specific for Type I fibers. Median $R^2 = 0.321$; 7 of 19 genes (37%) tested show essentially no correlation with *Myh4* expression.

DOI: https://doi.org/10.7554/eLife.34613.015

The following figure supplement is available for figure 3:

**Figure supplement 1.** Identifying skeletal muscle fiber type by *Myh* expression agrees with legacy data.

DOI: https://doi.org/10.7554/eLife.34613.016

noncoding RNA that regulates *Hox* gene expression in development (*Tsumagari et al., 2013*); as expected, it is expressed in all limb muscles (*Figure 4C*). In contrast, *Pitx2* is a transcription factor with a key role in driving extraocular muscle development (*Zhou et al., 2012*), and *Lhx2* is a transcription factor important for masseter muscle development (*Buckingham and Rigby, 2014*). The expression of both genes is consistent with regulatory activity that continues into adulthood and may contribute to maintaining proper mRNA expression patterns (*Figure 4D and E*). Notably, *Lhx2* is also involved in limb muscle development (*Hobert and Westphal, 2000*); it is of interest to determine how and why its expression is maintained in adult masseter but not adult limb tissues.

We speculate that computational modeling of transcription factor (TF) networks may resolve the mechanisms underlying tissue-specific mRNA profiles, particularly since neither developmental history nor fiber type composition are sufficient to explain the diversity of the skeletal muscle transcriptome. As a first step to this end, we used Ingenuity Pathway Analysis (IPA) (*Krämer et al., 2014*) to predict TFs upstream of differentially expressed genes among skeletal muscles. We then consolidated these predictions by restricting our analysis to TFs that are predicted to drive differential gene expression in at least five of the 11 total skeletal muscle tissues. A summary of these results is provided in *Table 2*, which includes 20 TFs predicted to contribute to skeletal muscle specialization. A simplified visual display of these data is provided in *Figure 5*, which highlights a complex web of 1) predicted upstream transcription factors, 2) numerous differentially expressed genes that we predict act as effectors of tissue specialization (including *Myostatin* and *Vegfa*), and 3) downstream processes involved in skeletal muscle disease and physiology. We note that *Smarca4* is predicted to be upstream of tissue-specific gene expression in nine of 11 tissues, and that it ranked as the most statistically confident prediction by IPA in four of these tissues. As *Smarca4* (*Brg1*) is involved in early transcriptional patterning of skeletal muscle (*Albini et al., 2015*), we believe it is a promising candidate for establishing transcriptional specialization in skeletal muscles.

The generation of tissue-specific promoter lines in transgenic mice will facilitate testing these candidates. Therefore, we identified transcripts expressed in a tissue-specific fashion in skeletal muscle. Plotting tissue specificity versus average expression reveals a bimodal distribution among all transcripts (*Figure 1—figure supplement 6A and B*). The majority of transcripts in skeletal muscle are expressed evenly across most skeletal muscle tissues, consistent with the notion that there is a minimal set of genes required to form sarcomeres and to generate contractile force. Nevertheless, a smaller set of transcripts (~5%) show nearly exclusive expression in a single skeletal muscle tissue. Manual examination of these tissue-specific genes reveals that many are found in head and neck muscles that have the most divergent developmental history of tissues in this study. Nonetheless, we found examples of tissue-specific genes in the diaphragm (*Figure 1—figure supplement 6C*) and the *FDB* (*Figure 1—figure supplement 6D*). A list of the most specific genes for any given muscle tissue is presented in *Supplementary file 6*. We note that one of the most specific genes for soleus, *Myh7*, has been validated previously using measures of protein expression (*Burkholder et al., 1994*). Nevertheless, we caution readers that independent validation using alternative approaches is recommended for the observations herein.

The presence of tissue-specific gene expression is consistent with a skeletal muscle transcriptome that is considerably more complex than previously appreciated. Moreover, the unprecedented depth

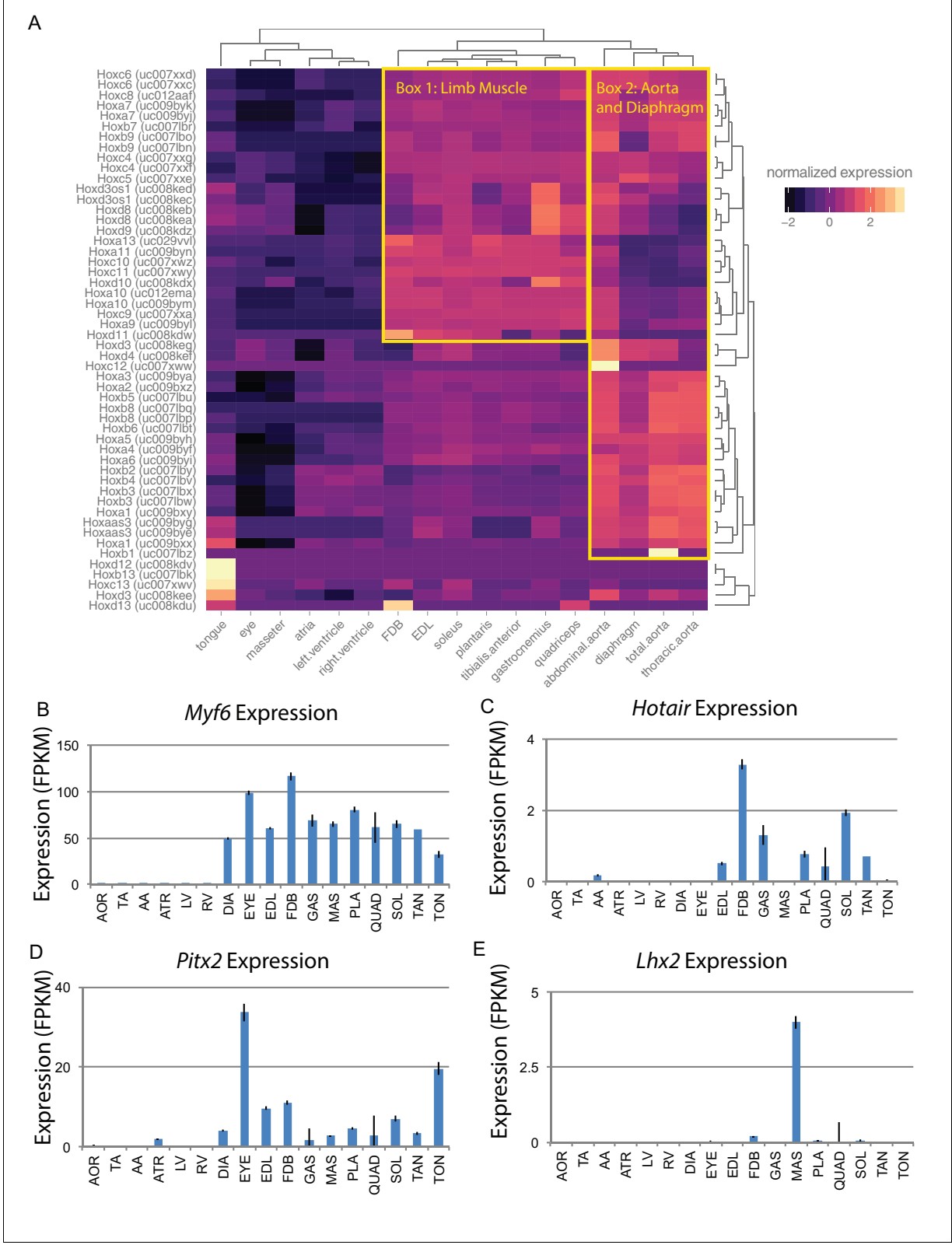

**Figure 4.** Developmental gene expression persists into adulthood in mouse skeletal muscle. (A) Normalized expression (Z-score by row) of all *Hox* gene transcripts is represented as a heat map. Row-normalization was chosen to display these data in a way that reveals the fine detail of all *Hox* genes, rather than those expressed at the highest levels. Overall similarity by *Hox* gene expression is represented as a dendrogram (top). One yellow box highlights a cluster of *Hox* genes expressed in limb skeletal muscle; a second yellow box highlights a cluster of *Hox* genes expressed in the aorta and

*Figure 4 continued on next page*

*Figure 4 continued*

diaphragm. (B) Expression of *Myf6* is shown as a bar graph. Muscle-specific expression of skeletal muscle differentiation factor persists into adulthood. (C) Expression of *Hotair*, a non-coding RNA involved in *Hox* gene regulation, is shown as a bar graph. *Hotair* expression is highly specific for a subset of skeletal muscle tissues involved in limb movement. (D) Expression of *Pitx2*, a gene involved in the development of head muscles, is shown as a bar graph. *Pitx2* expression is enriched in head and neck tissues such as the extraocular eye muscles and the tongue. (E) Expression of *Lhx2*, another gene involved in head and neck muscle development, is shown as a bar graph. *Lhx2* expression is highly specific for the masseter. AOR = total aorta, ATR = atria, DIA = diaphragm, EDL = extensor digitorum longus, EYE = extraocular eye muscles, FDB = flexor digitorum brevis, GAS = gastrocnemius, LV = left ventricle, MAS = masseter, PLA = plantaris, QUAD = quadriceps, RV = right ventricle, SOL = soleus, TA = thorascic aorta, TAN = tibialis anterior, TON = tongue. Error bars are ±S.E.M.

DOI: https://doi.org/10.7554/eLife.34613.017

**Table 2.** Transcription factors predicted to be upstream of differentially regulated genes.

| Activated TF | Top 25 TFs | Interacts with functionally significant muscle genes: |
|---|---|---|
| SP1 | 11 | *AR, Dmd, Foxo3, Hdac2, Hdac3, Hdac4, Hif1a, Kdm5a, Mef2c, Myh7, MylK, NFkB1, Rb1, Rxra, Tp53, Vegf, Vegfa* |
| HIF1A | 10 | *AR, Acta1, Acta2, Arnt2, Cnntb1, Epas1, Foxo3, Hdac2, Hdac3, Hdac4, Hdac5, Hdac7, Mef2c, Mstn, Myc, Myh1, Myh2, Myh3, Myh4, Myh6, Myl1, MylK2,NFkB (complex), Ppargc1a, Rb1, Rxra, Sp1, Tp53, Ttn, Ucp3, Vegf, Vegfa* |
| SMARCA4 | 9 | *Acta1, Acta2, Actb, Actl6a, Actn4, AR, Ctnnb1, Dysf, Gli1, Hdac2, Hdac4, Mef2c, Myc, Myh3, Myh7, Myh11, Myl1, Myl4, Myl5, MylK, Mylpf, MyoD1, MyoG, NFkB (complex), Pparg, Rb1, Rxra, Six1, Sp1, Tp53, Tpm1* |
| TCF7L2 | 9 | *AR, Ctnnb1, Epas1, Hdac2, Myc, MylK2, Myo6, Ppargc1a, Tp53* |
| ARNT2 | 8 | *Epas1, Gli1, Hif1a, Myh6, Myh7, Vegfa* |
| CTNNB1 | 8 | *Acta2, Actb, Actc1, Actn4, AR, Epas1, Foxo3, Gli1, Hdac2, Hdac3, Hdac4, Hdac5, Hdac7, Myc, Myf5, Myh3, Myh6, Myl4, MylK, MyoD1, MyoG, NFkB (complex), Rxra, Six1, Sp1, Stat5b, Smarca4, Tcf7l2, Tnnc1, Tp53, Vegfa* |
| STAT4 | 8 | Myc, NFkB (complex), Smarca4, Vegfa |
| MYC | 7 | *Acta1, Actb, Actn1, Actn4, Aimp2, AR, Ctnnb1, Egr2, Epas1, Foxo3, Gli1, Hdac2, Hdac3, Hdac5, Hif1a, Kdm5a, Myh7, Myl9, Mylpf, Myo1B, Myo1C, NFkB (complex), Rb1, Smarca4, Sp1, Stat4, Stat5b, Tcf7l2, Tnni3, Tnnt3, Tp53, Tpm1, Vegf, Vegfa* |
| TP53 | 7 | *Acta2, Actb, Actn1, Actn4, Aimp2, AR, Ctnnb1, Egr2, Epas1, Foxo3, Gli1, Hdac2, Hdac3, Hdac5, Hif1a, Myc, Myh9, Myh10, Myh16, Myl4, Myl9, MylK, Myo1c, Myo6, Myo10, MyoD1, Myof, NFkB (complex), Ppargc1a, Rb1, Smarca4, Sp1, Tcf7l2, Tp53, Tpm1, Tpm2, Tpm4, Ucp3, Vegf, Vegfa* |
| FOXO3 | 6 | AR, Ctnnb1, Egr2, Hdac2, Hif1a, Mef2c, Mstn, Myc, Myo6, MyoC, MyoD1, NFkB (complex), Ppargc1a, Rb1, Sp1, Tp53, Ucp2, Vegfa |
| EGR2 | 6 | *Foxo3, Mef2c, Myc, MyoC, Ucp3, Vegfa* |
| NFkB (complex) | 6 | *Actb, AR, Ctnn1b, Epas1, Hdac2, Hdac3, Hdac4, Hdac5, Hif1a, Mstn, Myc, MylK, MylK3, Myo1E, MyoD1, Stat4, Tp53, Vegfa* |
| RB1 | 6 | *Actb, Actc1, Actn2, Actn3, AR, Foxo3, Hdac2, Hdac3, Hif1a, Kdm5a, Mef2c, Myc, Mstn, Myh2, Myh4, Myh6, Myh7, Myh8, Myl1, Myl4, Myl6B, MyoD1, MyoM2, NFkB2, PPargc1a, Ryr1, Smarca4, Sp1, Tnnc1, Tnnc2, Tnni2, Tnnt1, Tp53, Tpm1, Tpm2, Vegfa* |
| GLI1 | 6 | *Arnt2, AR, Ctnnb1, Hdac2, Myc, Mef2c, Rxra, Smarca4, Tp53, Vegfa* |
| MEF2C | 6 | *Acta1, Actc1, Actn2, AR, Dmd, Egr2, Epas1, Foxo3, Gli1, Hdac3, Hdac4, Hdac5, Hdac7, Hif1a, Mstn, Myh1, Myh6, Myh7, Myl2, Myl4, Myl7, MylK2, Mylpf, MyoD1, MyoG, MyoM1, MyoM2, MyoT, MyoZ1, MyoZ2, Smarca4, Ppargc1a, Rb1, Sp1, Tnnc1, Tnni2, Tnnt2, Tpm1, Ttn, Vegfa* |
| STAT5B | 5 | *Actc1, AR, Ctnnb1, Myc, Myh1, Myh6, Myh7, Myl1, Myl2, Myl4, Myl7, NFkB (complex), Pparg, Rxra, Tp53, Tnnc1, Tnni1, Tnnt1, Tpm3* |
| PPARGC1A | 5 | *AR, Dmd, Epas1, Foxo3, Hdac5, Hif1a, Mef2c, Mstn, Myh6, Myh7, Myl2, MyoD1, MyoG, NFkB (complex), Rxra, Rb1, Tcf7l2, Tnni1, Tp53, Ucp2, Ucp3, Vegfa* |
| MYOD1 | 5 | *Acta1, Actc1, AR, Ctnnb1, Dmd, Dysf, Foxo3, Hdac3, Hdac4, Hdac5, Mef2c, Mstn, Myf5, Myf6, Myh2, Myh3, Myh7, Myl1, Myl4, Mylpf, Myo5b, NFkB1, Ppargc1a, Rb1, Rxra, Ryr1, Six1, Smarca4, Sp1, Tnnc1, Tnnc2, Tnni1, Tnni2, Tnnt1, Tnnt2, Tnnt3, Tp53, Tpm2, Ttn, Ucp3* |
| EPAS1 | 5 | AR, Arnt2, Ctnnb1, Hif1a, Mef2c, Myc, Myh4, Myo7a, MyoM2, NFkB (complex), Ppargc1a, Smarca4, Sp1, Tcf7l2, Tnni1, Tp53, Ucp2, Vegf, Vegfa |

| Inhibited TF | Top 25 TFs | Interacts with functionally significant muscle genes: |
|---|---|---|
| KDM5A | 8 | *Actc1, Actn2, Actn3, AR, Hdac2, Myc, Myh2, Myh4, Myh6, Myh7, Myh8, Myl1, Myl4, Myl6b, MyoM2, Rb1, Ryr1, Sp1, Tnnc1, Tnnc2, Tnni2, Tnnt1, Tnnt2, Tpm1, Tpm2, Vegf* |

DOI: https://doi.org/10.7554/eLife.34613.019

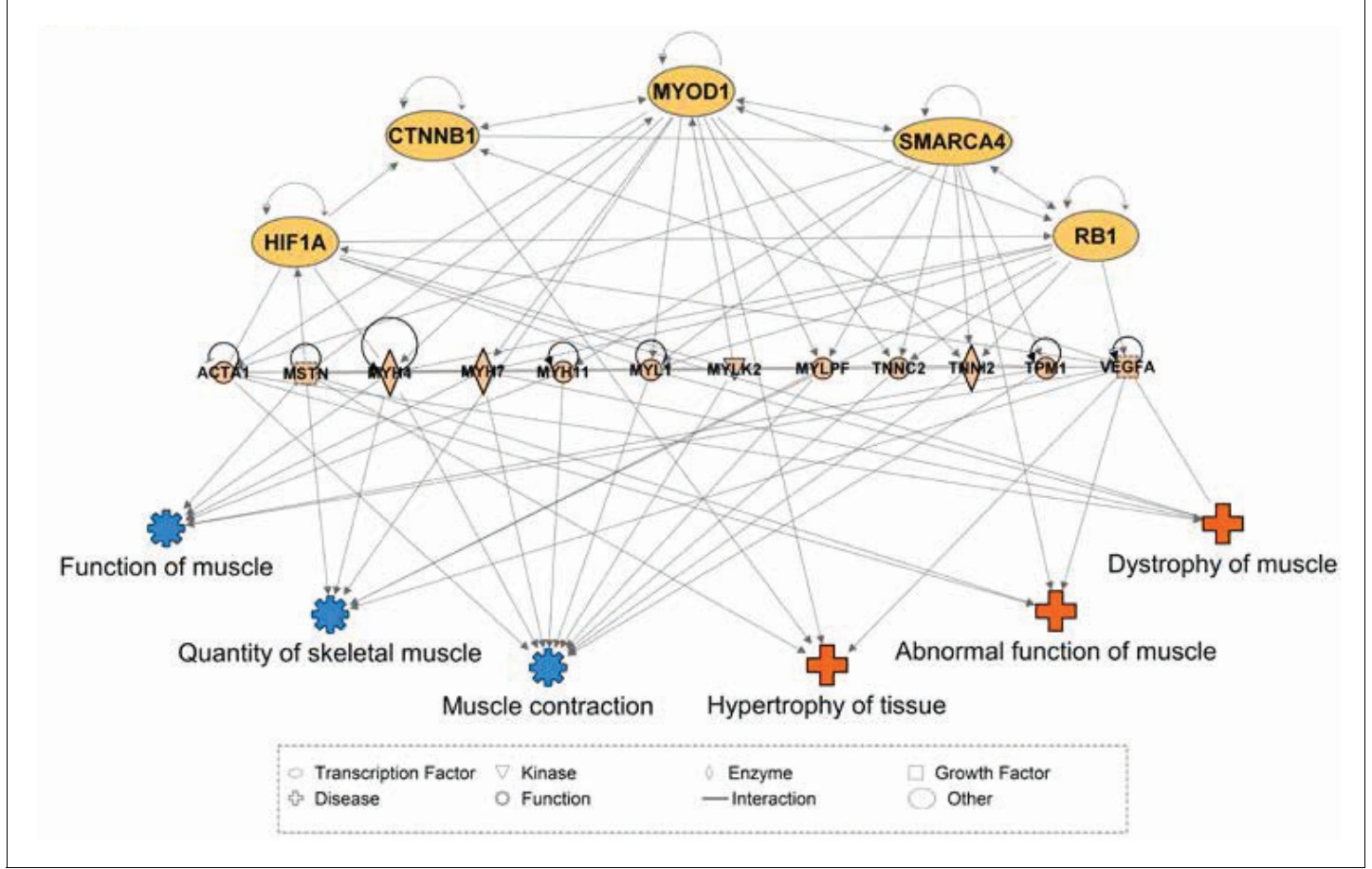

**Figure 5.** Network interactions of transcription factors predicted to be upstream of differentially expressed skeletal muscle genes. Ingenuity Pathway Analysis (IPA) was used to predict transcription factors upstream of differentially expressed muscle genes. The beige nodes (middle tier) represent muscle genes whose expression varies between muscle tissues. The orange nodes (top tier) are transcription factor genes predicted to contribute to muscle specific expression patterns. The blue and red nodes (bottom tier) represent biological functions and disease processes, respectively. Edges represent known, directional regulatory interactions. In the interest of clarity, this network has been manually trimmed to only include the most pertinent nodes. For a complete list of all predictions and regulated genes, please see *Table 2*.
DOI: https://doi.org/10.7554/eLife.34613.018

of sequencing coverage in muscle permits the discovery of heretofore-unannotated transcripts. Manual examination of 'gapped reads' spanning two or more loci in the genome indicates that hundreds to thousands of splicing events occur in muscle that have not been previously described (*Figure 1—figure supplement 7A*). The majority of these novel splicing events result in either inclusion of novel exons or the exclusion of exons previously thought to be constitutive (*Figure 1—figure supplement 7B*). To validate this observation, we designed oligonucleotide primers (*Figure 1—figure supplement 7C*) specific for two novel exons of *Myosin light chain kinase 4* (*Mylk4*), a gene with high expression (>200 FPKM) in many skeletal muscle tissues. PCR amplification and molecular cloning confirmed that these putative exons are included in full-length *Mylk4* transcripts in two different skeletal muscle tissues, *EDL* and *soleus* (*Figure 1—figure supplement 7D*). As the previously canonical splicing event linking exons 2 and 3 is detected by RNAseq in every muscle sample in this study, albeit at levels below the threshold for detection in RT-PCR, we conclude that transcripts including these putative exons are in actuality the predominant species of *Mylk4* mRNA. A list of all novel splice junctions is provided in *Supplementary file 7* and *8*.

There are significant differences in fiber type composition between analogous muscle tissues of different mammals. For example, mouse *soleus* is a mixture of fast- and slow fibers (*Figure 3A*), while rat *soleus* is almost entirely slow twitch (*Figure 6A*). Because of this, we asked whether gene

expression differences in mice are conserved across species. Expression of orthologous genes in mouse versus rat tissues has moderate levels of correlation ($R^2$ >0.6, *Figure 6B*), despite the difference in fiber type composition noted above. This reinforces the observation that tissue identity, rather than fiber-type composition, drives transcriptome diversity in muscle. Moreover, the vast majority of genes differentially expressed between *EDL* and *soleus* in both mice and rats changed in the same direction (*Figure 6C*). Taking this observation one step further, the fold change of all genes differentially expressed in mouse *EDL* compared to *soleus* (*Figure 6D*) is largely consistent with the fold change of same genes in rat *EDL* versus *soleus* (*Figure 6E*). Sex differences did not dramatically influence differential gene expression between *EDL* and *soleus* (*Figure 6—figure supplement 1*). Fewer than 3% of transcripts were differentially expressed between male and female rat *EDL* (2.7%) and male and female rat *soleus* (1.9%). Of these differentially expressed genes, most are up-regulated in males. This observation agrees with previous studies (*Roth et al., 2002*) and is consistent with the possibility that androgen response elements influence sex-specific gene expression differences in skeletal muscle. These results indicate that differentially expressed genes are largely conserved between mice and rats and suggest that these data may predict gene expression in related species.

Taken as a whole, there is considerable variance in gene expression profiles among skeletal muscle tissues, in stark disagreement with the assumptions of previous gene expression atlases. In the remainder of this paper, we will explore several analyses illustrating the utility of these data as resource for generating testable hypotheses related to tissue specialization.

Based on the likely conservation of gene expression patterns in humans, we speculate that genes associated with disease and those encoding drug targets will be of particular importance to follow-up studies. Of the roughly 23,000 genes encoded by the mouse genome, over 50% are differentially expressed among mouse skeletal muscle tissues. Of these, 3370 differentially expressed genes have human orthologs associated with disease, and 556 of those encode molecular targets of drugs on the market today (*Figure 7A*). These genes may contribute to the molecular mechanisms underlying differential disease susceptibility and pharmaceutical sensitivity in skeletal muscle tissues. As a resource for investigators, we provide a list of all differentially expressed genes specifically involved in human skeletal muscle disorders (*Supplementary file 9*).

To give one example of differential disease susceptibility, the aberrant expression of an embryonic isoform of *Pyruvate kinase* (*Pkm*) is involved in the mechanism of myotonic dystrophy (*Gao and Cooper, 2013*). Our data show that isoforms of *Pkm* are up-regulated by several standard deviations in *EDL* compared to all other muscle groups (*Figure 7B*). Myotonic dystrophy disrupts normal splicing and pathologically elevates *Pkm*, which in turn disrupts normal metabolism, decreasing oxygen consumption and increasing glucose consumption. Based on these observations, elevated expression of *Pkm* in adult tissues is hypothesized to be a critical step in the pathology of myotonic dystrophy (*Gao and Cooper, 2013*). Since *Pkm* is expressed much higher in *EDL* than all other muscle types (*Figure 7B*), we predict that *EDL* would be more sensitive to degeneration than other muscles. As myotonic dystrophy most dramatically affects certain subsets of muscle tissues, these observations suggest testable hypotheses regarding the underlying mechanism of disease susceptibility. Consistent with this possibility, we note with great interest that in mouse models, *EDL* is considerably more susceptible to muscle weakness than either diaphragm or *soleus* (*Moyer et al., 2011*).

In addition to disease susceptibility, these data may help explain differential drug sensitivity in muscle. To give one example, the drug chlorzoxazone (brand name: Lorzone) is used to treat muscle spasms. It is thought to act on the central nervous system (CNS) by regulating a potassium channel encoded by the gene *Kcnma1* (*Dong et al., 2006*). Although *Kcnma1* is expressed throughout the central nervous system (*ENCODE Project Consortium, 2012*), it is also found at comparable levels in most skeletal muscle tissues (*Figure 7C*). The two exceptions are extraocular eye muscles and the *soleus* that have greater than three-fold higher *Kcnma1* expression than other muscle tissues. The differential abundance of chlorozoxazone's target could influence either the efficacy of this drug or the severity of its side effects in different muscles. The expression in skeletal muscle of mRNAs encoding chlorzoxazone's protein target calls into question the assumption in the literature that this drug acts exclusively through the CNS. Moreover, given that some commonly prescribed drugs, such as glucocorticoids, cause muscle wasting in specific subsets of muscle tissues (*Schakman et al., 2008*), we speculate that this data set will be a valuable resource for exploring the mechanisms underlying differential drug sensitivity among skeletal muscles.

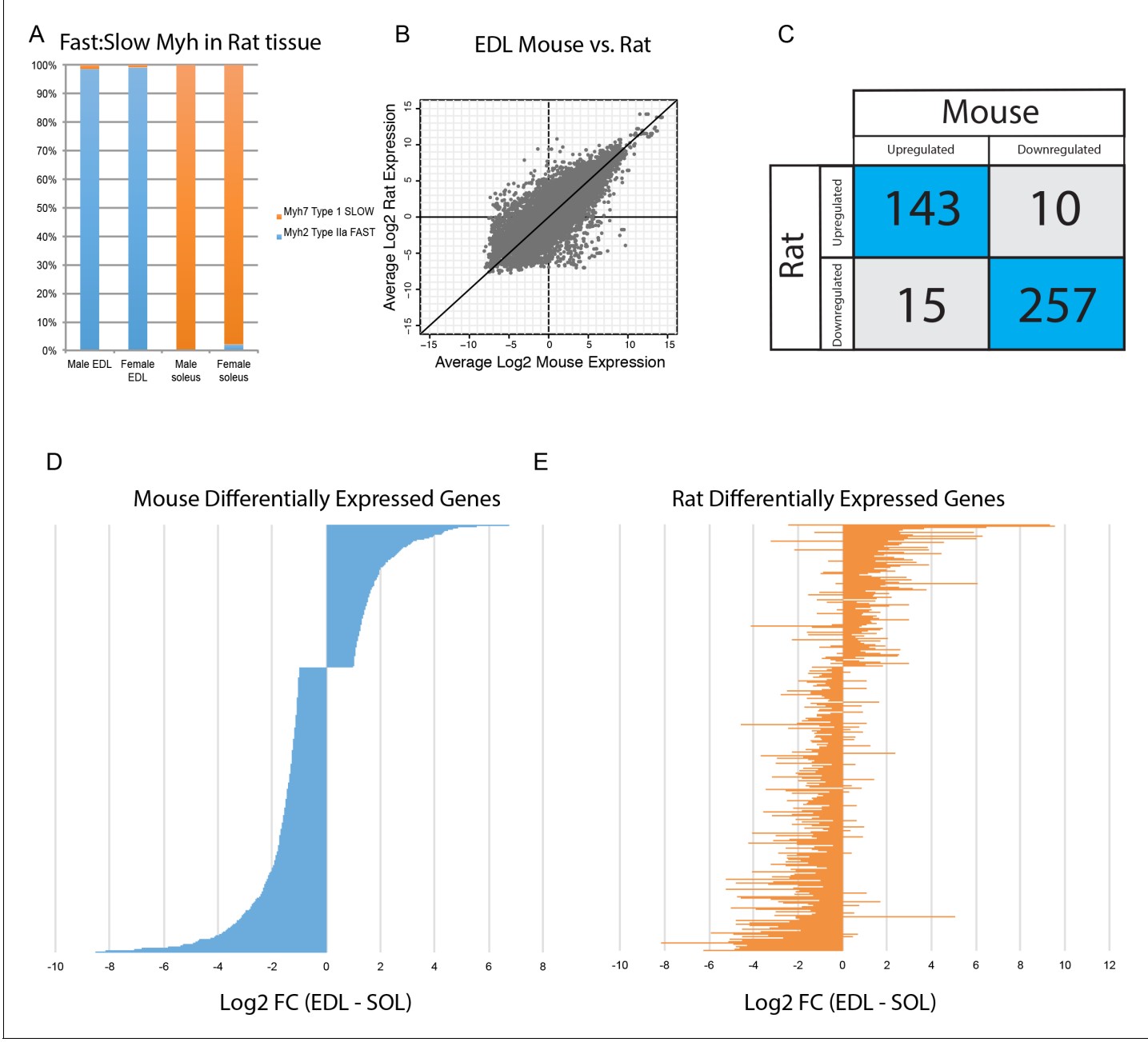

**Figure 6.** Differential gene expression is conserved between mice and rats. (**A**) The relative abundance of fast:slow *Myosin heavy chain* (*Myh*) transcripts in rat male and female *EDL* and *soleus* is shown as a bar graph. As expected, the fiber type composition of rat muscle is more homogeneous than mouse (compare with *Figure 3B*) (**B**) Scatter plot showing the overall similarity between mouse and rat transcriptomes in *EDL* ($R^2$ = 0.637). Each dot represents a single orthologous gene shared between mice and rats. Correlation between the transcriptomes of mouse and rat *soleus* is essentially the same as for *EDL* described above ($R^2$ = 0.662). (**C**) Of the 425 genes differentially expressed between *EDL* and *soleus* in both mice and rats, the majority (94%) were differentially expressed in the same direction (i.e., up-regulated in both mice and rats or down-regulated in both mice and rats). (**D**) The fold change for all rank-ordered differentially expressed genes in mice between EDL and soleus are plotted as a bar graph (N = 691, q < 0.05, fold change >2). (**E**) The fold change (*EDL/soleus*) for the rat orthologues of the genes in (**D**) are plotted as a bar graph. The order of genes in (**D**) and (**E**) is identical. The majority of genes in rat (>90%) show differential expression in the same direction as seen in mice (i.e., up-regulated in both mice and rats or down-regulated in both mice and rats).

DOI: https://doi.org/10.7554/eLife.34613.020

The following figure supplement is available for figure 6:

**Figure supplement 1.** Relatively few genes are differentially expressed between males and females; most of these are up-regulated in males.

DOI: https://doi.org/10.7554/eLife.34613.021

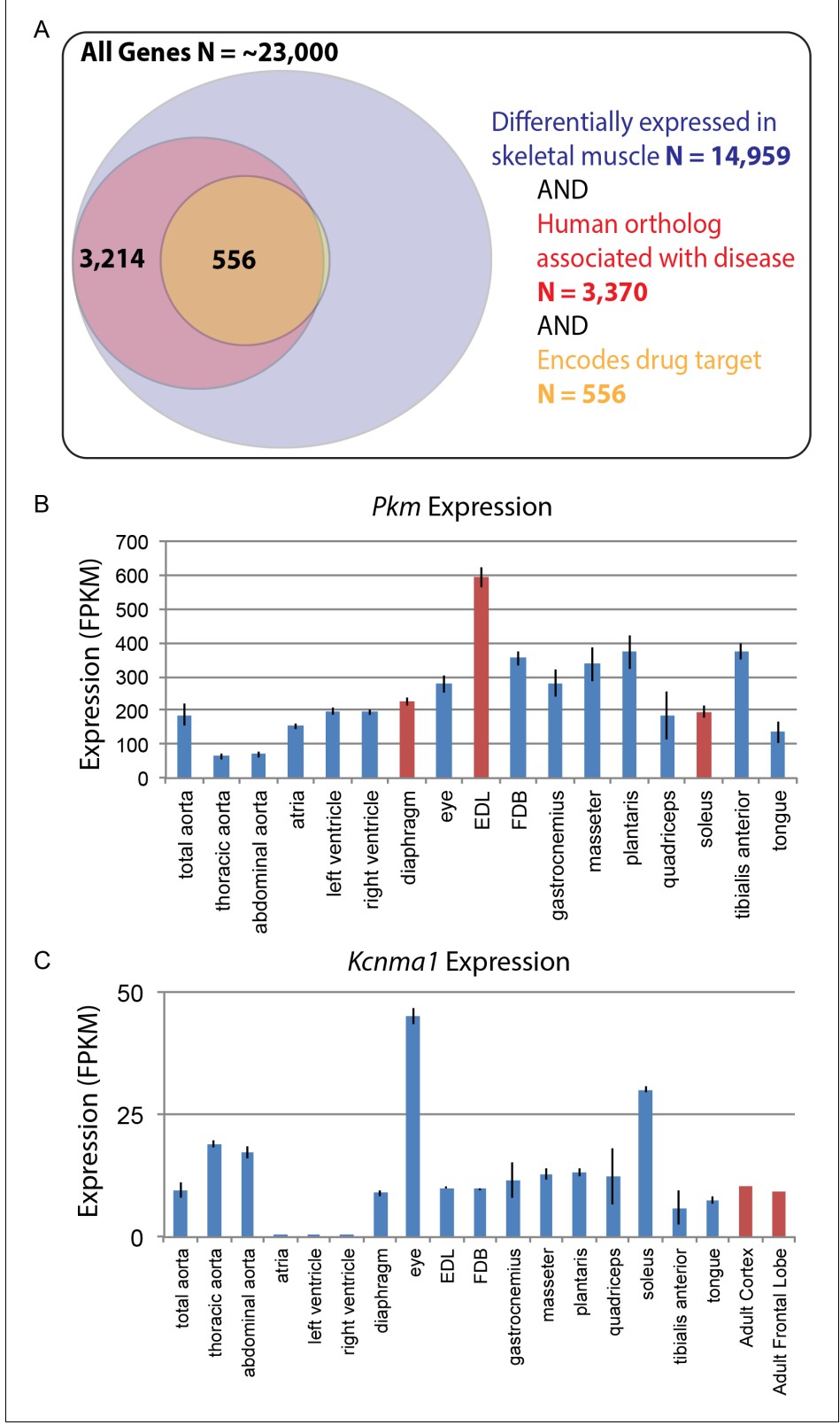

**Figure 7.** Differential gene expression in skeletal muscle may influence pathology and pharmacology. (**A**) Of roughly 23,000 genes in mice, 14,959 (~65%) are differentially expressed among skeletal muscle tissues (blue circle). Of these, 3370 (~15% of all genes) have orthologs known to influence human disease (red circle). Of these, 556 (>2% of all genes) encode the target of a marketed drug (orange circle). A small number of genes (N = 13)

*Figure 7 continued on next page*

*Figure 7 continued*

encode a drug target and are differentially expressed in skeletal muscle, but are not annotated as being associated with human disease, which explains why a sliver of the Venn diagram does not entirely overlap between red and orange circles. (B) The expression of *Pkm*, a key gene involved in the pathogenesis of myotonic dystrophy, is shown as a bar graph. Data points highlighted in red (*EDL*, *soleus*, and diaphragm) represent tissues that have been empirically tested for sensitivity to degeneration in a mouse model of myotonic dystrophy (***Moyer et al., 2011***). (C) Expression of *Kcnma1* is shown as a bar graph. *Kcnma1* encodes a potassium channel that is the molecular target of Chlorzoxazone, a drug prescribed as a muscle relaxant. Data points highlighted in red represent FPKM normalized gene expression in the adult cortex and adult frontal lobe as measured by the ENCODE consortium (***ENCODE Project Consortium, 2012***). Error bars are ±S.E.M., when available.

DOI: https://doi.org/10.7554/eLife.34613.022

The following figure supplements are available for figure 7:

**Figure supplement 1.** Skeletal muscle expresses numerous myokines.

DOI: https://doi.org/10.7554/eLife.34613.023

**Figure supplement 2.** Skeletal muscle expresses many genes involved in synapse assembly.

DOI: https://doi.org/10.7554/eLife.34613.024

Skeletal muscles are endocrine tissues, secreting numerous hormones that influence the physiology and metabolism throughout the body. Termed 'myokines', these signaling molecules are key regulators of human health and disease (***Lightfoot and Cooper, 2016***). The total number of myokines is currently unknown. Our data contribute to this field by comprehensively defining mRNA expression of candidate myokines for future study (***Figure 7—figure supplement 1*** and ***Supplementary file 10***). Dozens of genes encoding secreted proteins are differentially expressed among skeletal muscles at relatively high levels (N = 42 unique genes, FPKM >10). One notable example is *Vegfa* (q-value ~$10^{-30}$), a gene involved in angiogenesis, cardiac disease, cancer progression, and many other normal and pathological processes (***Smith et al., 2015***). Average expression of its predominant isoform is in the top 98th percentile of all transcripts expressed by skeletal muscle. Moreover, maximal expression of *Vegfa* in the diaphragm is nearly 10-fold greater than its minimal expression in the FDB, indicating that there are muscle-specific mechanisms for regulating *Vegfa* levels in particular and myokine levels in general. The serum concentration of VEGF-A in healthy adult humans is considerably greater than serum levels of IL6, a canonical myokine (***Lightfoot and Cooper, 2016***). Given that roughly 40% of the human body is comprised of skeletal muscle expressing high levels of *Vegfa*, we speculate that muscle may be a heretofore under-appreciated source of VEGF-A in circulation. Whether *Vegfa* expression by skeletal muscle has endocrine as well as paracrine functions is unknown.

Regenerative medicine has made considerable progress in generating skeletal muscle from stem cells (***Qazi et al., 2015***). Nevertheless, engineered tissues have important deficiencies in generating sufficient force and forming appropriate neuromuscular synapses (***Juhas and Bursac, 2013***). Given the extensive diversity observed among skeletal muscle tissues, we speculate that the inability to form proper synaptic connections in engineered tissue may be due to the expression of inappropriate or incomplete transcriptional programs. In other words, differentiating stem cells into generic skeletal muscle may not recapitulate all the cues necessary for muscle-specific synapse formation. Therefore, we examined the differential expression of genes implicated in synapse assembly (***Figure 7—figure supplement 2***). Dozens of transcripts involved in synapse formation are expressed at high levels (FPKM >10), suggesting that they are skeletal muscle mRNAs, rather than contamination from nearby neurons. One illustrative example, *Fbxo45*, is an E3 ligase involved in synapse formation. Mouse knock-outs of *Fbxo45* have disrupted neuromuscular junction (NMJ) formation in the diaphragm, resulting in early lethality (***Saiga et al., 2009***). Moreover, the *C. elegans* ortholog, *FSN-1*, also disrupts NMJ formation, with some synapses being over-developed while others are under-developed (***Liao et al., 2004***). We speculate that this protein and its orthologs may play a conserved role in tissue-specific NMJ assembly.

## Discussion

General textbook discussions of skeletal muscle typically focus on developmental patterning, neuro-muscular synapse physiology, or the biophysics of contractile functions (*Alberts, 2014*). This reflects the generally held belief that adult skeletal muscle is interesting only as a mechanical output of the nervous system. Functional genomics studies have acted on this assumption to the extent that every gene expression atlas generated to-date has selected at most one skeletal muscle as representative of the entire family of tissues. As such, the null hypothesis of this study was that gene expression profiles would be largely similar among skeletal muscle tissues.

However, as more than 50% of transcripts are differentially expressed among skeletal muscles (*Figure 1C*), and 13% of transcripts are differentially expressed between any two skeletal muscle tissues on average (*Figure 1E*), the data are entirely inconsistent with the null hypothesis. These results indicate that there is no such thing as a representative skeletal muscle tissue. Instead, skeletal muscle should be viewed as a family of related tissues with a common contractile function but widely divergent physiology, metabolism, morphology, and developmental history. Based on these antecedents, it comes as no surprise that the transcriptional programs maintaining skeletal muscle specialization in adults are highly divergent as well.

This study is the first systematic examination of transcriptome diversity in skeletal muscle. At greater than 200 million aligned short nucleotide reads per tissue and six biological replicates apiece, this data set is unprecedented in its scope, accuracy, and reproducibility (*Figure 1—figure supplement 4*, *Figure 1—figure supplement 4*, and *Figure 3—figure supplement 1*). Moreover, the depth of sequencing allows the detection of previously unannotated transcripts that may play a role in muscle physiology (*Figure 1—figure supplement 2* and *Figure 1—figure supplement 7*, *Supplementary file 7* and *8*).

Besides establishing that skeletal muscles have considerable differences in their transcriptomes, the key significance of this paper will be as a resource for future studies. Therefore, we have made our analyzed data freely available (http://muscledb.org), and all raw data may be downloaded from NCBI's GEO. This resource will allow investigators to perform analyses beyond the scope of this paper, such as generating muscle-specific *Cre*-recombinase mouse strains for genetically manipulating specific muscle groups. Most importantly, these data will provide the foundation for computational modeling of transcription factor networks, a method we believe will uncover the genetic mechanisms that establish and maintain muscle specialization. To this end, we have used principal component analysis (*Figure 2*) and related approaches (*Figure 1—figure supplement 6* and *Figure 1—figure supplement 7*) to show that expression of key skeletal muscle genes including different versions of *Troponin*, *Tropomyosin*, and *Calsequestrin* are highly predictive of skeletal muscle identity. Moreover, we used pathway analysis (*Figure 5* and *Table 2*) to identify 20 candidate transcription factors that may drive transcriptional specialization in muscle cells.

One potential criticism is that gene expression does not always predict protein level and therefore function. We acknowledge that many processes besides steady-state mRNA levels regulate protein expression. Nevertheless, regulation of mRNA expression is unquestionably of biological importance in muscle cells, and transcriptional profiling predicts the majority of protein expression levels even in highly dynamic settings (*Robles et al., 2014*). Moreover, the larger dynamic range of RNAseq measurements permits a more comprehensive description of expression profiles than would be possible with existing proteomic technology. We look forward to proteomic studies making use of these mRNA data, especially the identification of novel spliceforms, to generate improved catalogs of protein expression in skeletal muscle.

We further acknowledge that the samples collected and analyzed in this study are bulk tissues rather than single-fiber preparations, and that contaminating tissues such as vasculature and immune cells may influence some gene expression measurements. However, our data agrees with the few single-fiber profiling papers in the literature (*Chemello et al., 2011*), indicating that this potential bias is unlikely to confound the major observations herein. Consistent with this, the genes whose expression is most predictive of skeletal muscle identity are almost unanimously canonical skeletal muscle genes (*Supplementary file 3*). Furthermore, we emphasize that the majority of studies in the field use bulk tissues rather than single cell preparations. As such, our experimental design yields the greatest possible consistency with previous and future studies. In short, we believe whole tissue

expression profiling provides a critical reference point and the rationale for follow-up work examining single-fiber gene expression.

Acquired and genetic diseases show remarkable selectivity in which muscles they affect and which muscles they spare. For example, Duchenne muscular dystrophy severely affects the diaphragm and proximal limb extensors, while oculopharyngeal dystrophy causes weakness in the neck, facial, and extraocular muscles (*Emery, 2002*). At present, there is no satisfying explanation for how this occurs. A reasonable hypothesis is that intrinsic properties of muscle cells, such as gene expression, determine their sensitivity to different pathological mechanisms. These data are a starting point for future studies on how specialized transcriptional programs in muscles are maintained and how they ultimately influence disease. As an illustrative example, we note that the naturally elevated expression of *Pkm* in *EDL* may in part explain how this muscle is most dramatically affected in mouse models of myotonic dystrophy (*Figure 7*).

Finally, skeletal muscle is an endocrine organ that regulates many normal and pathological processes, including sleep, bone health, diabetes, cancer, and cardiovascular disease (*Giudice and Taylor, 2017*; *Iizuka et al., 2014*; *Karsenty and Olson, 2016*). At roughly 40% of an adult human's body weight, skeletal muscle has an enormous capacity to influence other tissues through the expression of local or systemic signaling molecules. This study reveals extensive differential expression of putative myokines with largely unexplored functional significance (*Figure 7—figure supplement 1* and *Supplementary file 10*). As such, we predict these data will be instrumental in future studies of the endocrine mechanisms through which skeletal muscle regulates health and disease.

## Materials and methods

### Animal care and tissue collection

Adult male *C57Bl6J* mice were acquired from Jackson Laboratories at ten weeks of age. They were housed in light tight cages with a 12L:12D light schedule for four weeks with water and normal chow *ad libitum*. At 14 weeks of age, mice were sacrificed at between two and five hours after lights-on (i.e. ZT 2–5), and muscle tissues were rapidly dissected and flash frozen in liquid nitrogen for subsequent purification of total RNA. Adult male and female *Sprague Dawley* rats were obtained from Charles River (Wilmington, MA) at 12 weeks of age. Rats were housed in pairs with a 12 hr:12 hr light/dark cycle, and standard rat chow and water were provided *ad libitum*. At 14 weeks of age, rats were sacrificed at between two and five hours after lights-on (i.e. ZT 2–5), and muscle tissues were rapidly dissected and flash frozen in liquid nitrogen for subsequent purification of total RNA. Three animals were sacrificed per biological replicate. All animal procedures were conducted in compliance with the guidelines of the Association for Assessment and Accreditation of Laboratory Animal Care (AAALAC) and were approved by the Institutional Animal Care and Use Committee at University of Kentucky.

### RNA purification and library preparation

Between 5–20 mg of frozen tissue were manually homogenized in Trizol reagent (Invitrogen), and total RNA was purified using a standard chloroform extraction. RNA samples for gene expression analysis were mixed with an equal volume of 70% ethanol and further purified with RNEasy columns (Qiagen, Germany) using the manufacturer's protocol. RNA was purified from tissue collected from individual mice; samples from the three individual mice of each biological replicate were then pooled together in equimolar amounts for further analysis.

To assess RNA integrity, aliquots of each sample were denatured for 2 min at 70°C and analyzed on the Agilent 2100 Bioanalyzer using Eukaryote Total RNA Nano chips according to manufacturer's protocol. RNA integrity numbers (RINs) for all samples were above 8.0 with a median RIN of 9.2. Libraries were prepared using the Illumina Truseq Stranded mRNA LT kit using single end indexes according to manufacturer's protocol. Approximately 500 ng of total RNA was used as starting material and amplified with 13 cycles of PCR. Libraries were validated for size and purity on the Agilent 2100 Bioanalyzer using DNA 1000 chips according to manufacturer's protocol.

## RNAsequencing and analysis

Pilot runs to verify library integrity were sequenced on an Illumina MiSeq (University of Missouri—St. Louis), and subsequent sequencing was performed on an Illumina HiSeq 2500 (University of Michigan) or HiSeq 3000 (Washington University in St. Louis). All sequenced reads were 50 bp, single-end. Raw reads were aligned to the genome and transcriptome of *Mus musculus* (build mm10) or *Rattus norvegius* (build rn5) using RNAseq Unified Mapper (RUM) (*Grant et al., 2011*) with the following parameters: '*–strand-specific –variable-length-reads –bowtie-nu-limit 10 –nu-limit 10*'. Mouse and rat gene models for RUM alignments were based on UCSC gene models updated as of December 2014. 65%–92% of reads uniquely mapped to the genome/transcriptome, and the total number of aligned reads (including unique and non-uniquely aligned reads) was at least 94.7% for every replicate sample (*Supplementary file 1*). FPKM values for each transcript/exon/intron were calculated by RUM using strand-specific unique reads normalized to the total number of reads uniquely aligned to the nuclear genome. To account for variability in the mitochondrial content of different muscles, uniquely aligned mitochondrial reads were excluded from the denominator. R-squared values of log-transformed FPKMs between biological replicates of the same tissue were generally ~0.93 (*Figure 1—figure supplement 3*), and internal controls (e.g. *Figure 1—figure supplement 4*) were used to verify the biological validity of these measurements.

Differential expression between tissues was determined by one-way ANOVA of log-transformed FPKM values and adjusted for multiple testing using a Benjamini-Hochberg q-value (*Hochberg and Benjamini, 1990*). Unless otherwise noted, a q-value less than 0.01 and fold-change greater than 2.0 among transcripts expressed with an FPKM > 1.0 was deemed statistically significant. Transcripts were deemed to be expressed at FPKM > 1.0, unless otherwise noted. Disease and drug target associations were identified using public data sets (DrugBank v5.0.9 and DisGeNET or Ingenuity Pathway Analysis). Mouse genes were mapped to human orthologs using the BiomaRt package for R (*Durinck et al., 2009*). Dendrograms were produced in R. A distance matrix was calculated for all (*Figure 1D*) or subsets (*Figures 3B* and *4A*) of transcripts using the dist function using the standard options (Euclidean distance; *Figure 1—figure supplement 5*). Hierarchical clustering was performed using the hclust function using the complete agglomeration method. The code used to produce the heatmaps and dendrograms is freely available on GitHub (https://github.com/flaneuse/muscleDB [*Hughes, 2017*]; copy archived at https://github.com/elifesciences-publications/muscleDB).

For principal component analysis, log transformed FPKM values of expressed transcripts were passed to the bootPCA function from the bootSVD R library. A transcript was considered expressed if its mean log2(FPKM +1) value across replicates was >1.0 for at least one tissue. To estimate sampling variability, 10,000 bootstrap replicates were generated, and 99% confidence intervals computed for the relevant statistical functionals (*Fisher et al., 2016*). Component loadings onto PC1 (i. e., the correlation between expression levels and PC1) were computed using R's cor function with default options.

The PC plot was created using the autoplot function from the package ggfortify.

Ingenuity Pathway Analysis (v.43605602) was used to evaluate major transcription factor networks involved in the observed transcriptional variation between skeletal muscle tissues (*Krämer et al., 2014*). Lists of differentially upregulated transcripts were downloaded from pairwise comparisons within MuscleDB (N = 110; FC > 2; q < 0.01). Each list was analyzed using log2 fold change in IPA's core analysis feature. If the list of upregulated genes contained more than 3000 genes, the q-value corresponding to the 3000th gene was used as an alternative threshold. Comparison analyses were run in IPA by grouping the core analyses for a single tissue (N = 11). The upstream analysis tool within the comparison analysis inferred potential transcriptional regulators based on the gene expression patterns within and between samples for a single tissue. The top 25 transcription factors consistent with the pattern of upregulation were recorded and the 20 most common TFs across all tissues were compiled into a table. The interactions shown were derived from the known interactions listed on Ingenuity Knowledge Base's (IKB) summary pages for each TF. The network shown was constructed using IPA's pathway designer. The regulated genes in the network were included if two independent pairwise comparisons (using four unique tissues) showed differential expression in MuscleDB (q < 0.01; FC > 1.7). The diseases and functions were added using IPA's data overlay

tool, which is based on the known interactions in the IKB. For the sake of clarity, only diseases and functions highly related to skeletal muscle physiology were included in *Figure 5*.

Using the same gene lists described above, we used DAVID Bioinformatics Resources 6.8 for functional GO analysis (*Huang et al., 2009a*; *Huang et al., 2009b*). To generate an appropriate background list for analysis, genes expressed in at least one of eleven skeletal muscle tissues were selected and converted into ENSEMBL gene IDs using either an ENSEMBL annotation file or the DAVID ID conversion tool. Genes with ambiguous accessions during conversion were removed. Enriched GO terms were filtered using a fold enrichment threshold of 2 and a false-discovery threshold of 0.05. For specific tissues, GO terms enriched in pairwise comparisons with ten other skeletal tissues were pooled to determine the most enriched GO terms.

We used package WGCNA 1.63 in R version 3.3.3 to detect trait related modules using the 'automatic network construction and module detection' method with a soft-thresholding power of 18 and a minimum module size of 30 (*Langfelder and Horvath, 2008*; *Zhang and Horvath, 2005*). Transcripts expressed in at least one of all 17 muscle tissues were collected as input. External traits were specific tissues or one of the four muscle categories identified in *Figure 1E*. For modules of interest, transcripts were converted from either Refseq or UCSC IDs to gene symbols using the biotools.fr online converter.

## Validation and novel splicing events

Independent biological replicates of mouse *EDL* and *soleus* tissues were collected as described above. Total RNA was purified and integrity was verified as described above. Reverse transcription reactions were performed with 500 ng starting material using the manufacturer's protocol (TaqMan Fast Universal PCR Master Mix, Applied Biosystems). qPCR was performed with ABI Taqman probes on a Stratagene MX3005 instrument (Pcp4l1:mm01295270_m1, Fam129a:mm00452065_m1, Fhl2, mm00515781_m1, Tsga10:mm01228282_m1, Stau2:mm00491782_m1, Prkag3:mm00463997_m1, Mstn:mm01254559_m1, Plcd4:mm00455768_m1, Myl1:mm00659043_m1, Igfbp5:mm00516037_m1, Ipo8:mm01255158_m1) using the manufacturer's recommendations.

To identify novel splicing events, previously unknown introns were identified from the junctions_all.rum file in RUM's output. These results were then sorted by the number of reads aligning to that splice junction and manually curated for follow-up studies. To validate novel splicing events, PCR reactions were performed using 1 ul of the reverse transcription reaction using the manufacturer's protocol (Clontech Takara PCR kit). PCR products were visualized on a 1% agarose gel using conventional methods. The primary PCR products for *EDL* and *soleus* (*Figure 1—figure supplement 7D*) from primers Exon 2- > Exon 2.1 and Exon2 - > Exon3 were excised, purified, and TOPO cloned using the manufacturer's protocol (TOPO TA, Thermo Fisher). Cloned fragments were sequenced using conventional Sanger methods. All PCR primers were ordered from IDT; primer locations are described in *Figure 1—figure supplement 7C*. Primer sequences as follows: Exon 2 – AGGATC TCAGATTTGCTCACG, Exon 2.1 – GGATCCACTTTCCAGAATGC, Exon 2.2 – CATCTTTGCACC TGCATTC, Exon 3 – TATGGTCCAACCGTGCACTA, CtrlF – AGTGTGGGCGTCATCACAT, CtrlR – G TGGAGCTTGTGGTCTGACA.

## Immunohistochemistry

Validation experiments using immunohistochemistry were performed for *Figure 1—figure supplement 6* using the following antibodies: UPK1B polyclonal antibody 1:200 (PAB25730, Abnova) and BBOX1 polyclonal antibody 1:400 (NBP1-32327, Novus Biologicals). We note that neither antibody is well characterized for immunostains in any tissue, and that our experiments revealed no specific staining.

## Data availability

All raw data and. bed files are available on NCBI's Gene Expression Omnibus (accession number: GSE100505), and transcript-level expression values can also be downloaded from MuscleDB (http://muscledb.org/), a web application built using the ExpressionDB platform (*Hughes et al., 2017*).

## Acknowledgements

We thank Jeanne Geskes, Robert Lyons, and the University of Michigan DNA sequencing core facility for assistance with next-generation sequencing. We thank Ron Anafi, (UPenn), Jeff Haspel (Washington University Department of Medicine), John Hogenesch (Cincinnati Children's Hospital), Patty Parker (UMSL), Bob Ricklefs (UMSL), and members of the Hughes and Esser laboratories for helpful discussion and technical support throughout this project. We thank Dr. Michael Chicoine (Washington University Department of Neurosurgery) for inestimable contributions without which this paper would never have been written. Work in the Gong lab is supported by NIH award HL106843. Work in the Esser lab is supported by NIH award R01AR066082. Work in the Hughes Lab is supported by NIH award R21AR069266 and start-up funds from the Department of Medicine at Washington University in St. Louis. We thank the Genome Technology Access Center in the Department of Genetics at Washington University School of Medicine for help with genomic analysis. The Center is partially supported by NCI Cancer Center Support Grant #P30 CA91842 to the Siteman Cancer Center and by ICTS/CTSA Grant# UL1 TR000448 from the National Center for Research Resources (NCRR), a component of the National Institutes of Health (NIH), and NIH Roadmap for Medical Research. This publication is solely the responsibility of the authors and does not necessarily represent the official view of NCRR or NIH.

## Additional information

### Funding

| Funder | Grant reference number | Author |
| --- | --- | --- |
| National Institute of Arthritis and Musculoskeletal and Skin Diseases | AR069266 | Karyn A Esser<br>Michael E Hughes |
| National Institute of Arthritis and Musculoskeletal and Skin Diseases | AR066082 | Karyn A Esser |

The funders had no role in study design, data collection and interpretation, or the decision to submit the work for publication.

### Author contributions

Erin E Terry, Resources, Data curation, Formal analysis, Validation, Investigation, Visualization, Methodology, Project administration, Writing—review and editing; Xiping Zhang, Lance A Riley, Resources, Investigation, Methodology, Writing—review and editing; Christy Hoffmann, Jiajia Li, Resources, Data curation, Formal analysis, Validation, Investigation, Visualization, Methodology, Writing—review and editing; Laura D Hughes, Resources, Data curation, Software, Formal analysis, Validation, Investigation, Visualization, Methodology, Writing—review and editing; Scott A Lewis, Data curation, Software, Formal analysis, Investigation, Visualization, Methodology, Writing—review and editing; Matthew J Wallace, Formal analysis, Validation, Investigation, Visualization, Methodology, Writing—review and editing; Collin M Douglas, Miguel A Gutierrez-Monreal, Investigation, Methodology; Nicholas F Lahens, Conceptualization, Formal analysis, Investigation, Methodology, Writing—review and editing; Ming C Gong, Conceptualization, Supervision, Methodology, Writing—review and editing; Francisco Andrade, Conceptualization, Supervision, Funding acquisition, Methodology, Writing—review and editing; Karyn A Esser, Conceptualization, Resources, Data curation, Supervision, Funding acquisition, Validation, Investigation, Methodology, Project administration, Writing—review and editing; Michael E Hughes, Conceptualization, Resources, Data curation, Formal analysis, Supervision, Funding acquisition, Validation, Investigation, Visualization, Methodology, Writing—original draft, Project administration, Writing—review and editing

### Author ORCIDs

Erin E Terry http://orcid.org/0000-0002-1334-4238
Nicholas F Lahens http://orcid.org/0000-0002-3965-5624

Karyn A Esser (ID) https://orcid.org/0000-0002-5791-1441
Michael E Hughes (ID) http://orcid.org/0000-0002-8828-3732

## Ethics

Animal experimentation: All animal procedures were conducted in compliance with the guidelines of the Association for Assessment and Accreditation of Laboratory Animal Care (AAALAC) and were approved by the Institutional Animal Care and Use Committee at University of Kentucky (IACUC assurance number: A-3336-01).

## Decision letter and Author response

Decision letter https://doi.org/10.7554/eLife.34613.039
Author response https://doi.org/10.7554/eLife.34613.040

# Additional files

### Supplementary files

• Supplementary file 1. Details on the total number of reads and aligned reads for every replicate sample.
DOI: https://doi.org/10.7554/eLife.34613.025

• Supplementary file 2. All transcripts differentially expressed among skeletal muscles. This table uses a statistical cut-off of FDR < $10^{-6}$ for one-way ANOVAs among mouse skeletal muscles and minimum expression >1.0 FPKM in at least one tissue. Alternative filtering options are available on http://muscledb.org.
DOI: https://doi.org/10.7554/eLife.34613.026

• Supplementary file 3. Top 100 genes with the highest correlation with PC1. This Table expands on *Figure 2D*.
DOI: https://doi.org/10.7554/eLife.34613.027

• Supplementary file 4. Genes comprising the most statistically significant modules in WGCNA analysis for smooth, cardiac, skeletal muscle cluster 1, and skeletal muscle cluster 2. Modules with the highest positive correlation with tissue identity were selected. The number of genes in each module is variable.
DOI: https://doi.org/10.7554/eLife.34613.028

• Supplementary file 5. Genes comprising the most statistically significant modules in WGCNA analysis for all skeletal muscles. Modules with the highest positive correlation with tissue identity were selected. The number of genes in each module is variable. We note that not every skeletal muscle tissue had a statistically significant (p<0.01) module.
DOI: https://doi.org/10.7554/eLife.34613.029

• Supplementary file 6. Genes with the highest specificity index in every tissue.
DOI: https://doi.org/10.7554/eLife.34613.030

• Supplementary file 7. Comprehensive list of all novel splicing events detected in mouse skeletal muscle and the muscles in which they are found. At least 10 reads spanning the novel exon:exon junction needed to be observed for inclusion in this list. All *.bed files are available on NCBI's GEO.
DOI: https://doi.org/10.7554/eLife.34613.031

• Supplementary file 8. Comprehensive list of all novel splicing events detected in rat skeletal muscle and the muscles in which they are found. At least 10 reads spanning the novel exon:exon junction needed to be observed for inclusion in this list. All *.bed files are available on NCBI's GEO.
DOI: https://doi.org/10.7554/eLife.34613.032

• Supplementary file 9. Differentially expressed genes annotated as being involved in skeletal muscle disease.
DOI: https://doi.org/10.7554/eLife.34613.033

• Supplementary file 10. Every unique gene that is a candidate myokine. This Table is distinct from *Figure 7—figure supplement 1* in that no filtering based on differential expression was performed, and the filter for minimal expression is less stringent (i.e. FPKM > 1 instead of FPKM > 10).

DOI: https://doi.org/10.7554/eLife.34613.034

• Transparent reporting form
DOI: https://doi.org/10.7554/eLife.34613.035

## Data availability

RNA Sequencing data have been deposited in GEO under accession code GSE100505. Analyzed data are available on http://muscledb.org.

The following dataset was generated:

| Author(s) | Year | Dataset title | Dataset URL | Database, license, and accessibility information |
|---|---|---|---|---|
| Terry EE, Zhang X, Hoffmann C, Hughes LD, Lewis SA, Li J, Wallace MJ, Riley LA, Douglas CM, Gutierrez-Montreal MA, Lahens NF, Gong M, Andrade F, Esser KA, Hughes ME | 2018 | Transcriptional Profiling Reveals Extraordinary Diversity Among Skeletal Muscle Tissues | http://www.ncbi.nlm.nih.gov/geo/query/acc.cgi?acc=GSE100505 | Publicly available at the NCBI Gene Expression Omnibus (accession no: GSE100505) |

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
