## [Decision Letter]

[Editors' note: the authors’ plan for revisions was approved and the authors made a formal revised submission.]

Thank you for sending your article entitled "Transcriptional profiling reveals extraordinary diversity among skeletal muscle tissues" for peer review at *eLife*. Your article has been favorably evaluated by Didier Stainier (Senior Editor) and three reviewers, one of whom is a member of our Board of Reviewing Editors.

Given the list of essential revisions, including new experiments, the editors and reviewers invite you to respond within the next two weeks with an action plan and timetable for the completion of the additional work. We plan to share your responses with the reviewers and then issue a binding recommendation.

In the current manuscript entitled "Transcriptional profiling reveals extraordinary diversity among skeletal muscle tissues", the authors addressed an important and outstanding issue in the field of skeletal muscle research: how similar or different are the tissues from distinct skeletal muscle groups?

Based on their analysis, the authors concluded that distinct skeletal muscle groups are profoundly different in their gene expression profiles and provided some useful and novel examples to understand these differences. Overall, the current manuscript is of high relevance and importance to the skeletal muscle field. However, as a Resource paper, it may need to develop or implement new technologies/computational models and potential validation experiments to be of high-level interest to the broader audience. The data suggests that neither developmental history nor fiber composition could explain transcriptome diversity among different muscles and the question was left unanswered-based on the dataset it is possible to extract correlates that could explain the diversity.

A major strength of this manuscript is that it assessed this issue on a comprehensive level.

A major concern is whether this manuscript is more appropriately suited in a specialized journal.

Major issues:

1) Due to the tissue harvesting procedures and the nature of bulk RNA-Seq used in this manuscript, it is likely that at least part of the differences observed among distinct skeletal muscle groups could be due to the other cell types present in the muscle tissues rather than from the skeletal muscles per se. Confirmation in MSCs/FAPs and endothelial cells versus purified single fibers from EDL and FDB would provide support.

2) A considerable number of conclusions are solely based on the RNA-Seq data without further experimental validation. For example, in Figure 4 the authors showed two examples of genes that can distinguish DIA and FDB, respectively, from all the other skeletal muscle types examined. The authors should also provide additional evidence to confirm their findings, such as RNA in situ hybridization or tissue IHC. As unique skeletal muscle type markers can be very valuable to the field, I suggest the authors to perform a more systematic analysis to find unique markers for all the skeletal muscle types in their study, and back up some of the markers using in situ or IHC as suggested above.

3) To obtain a better overview of transcriptional differences among muscle groups, it is advisable to perform GO analysis to examine what functionally-related gene sets are preferentially enriched among distinct skeletal muscle groups.

4) As transcription factors (TFs) commonly serve as nodes of gene expression regulation, it is recommended to perform TF network analysis to examine major TFs/TF networks that drive the transcriptional differences among muscle groups.

5) WGCNA analysis can be performed to examine the major and unique sets of genes that define each muscle groups. It can be helpful to include also smooth and cardiac muscles as an internal control for the analysis.

6) Try to identify molecular correlates of muscle diversity.

7) As a Resource paper, the authors should list all the differentially expressed genes and novel splicing events in an excel spreadsheet.

*Reviewer #1:*

The manuscript by Terry et al. investigates the transcriptional diversity among different muscles, including skeletal, smooth and cardiac muscle tissues from different species. Using RNA-sequencing, the authors assembled a dataset, termed MuscleDB, to reveal the transcriptional diversity of muscle tissues. Overall, the manuscript has good experimental design and quality control to ensure data accuracy and reproducibility (e.g. high replicate number for RNA-seq experiments, adequate sequencing depth and data validation). The work presents a new dataset to fill the knowledge gap regarding to whole muscle diversity. However, the data suggests that neither developmental history nor fiber composition could be the explanation of transcriptome diversity among different muscles and the question was left unanswered. Also, as mRNA levels are measured in a whole muscle context, the underlying complexity of different cell types within different muscles can affect the result of RNA-seq profiling. For instance, the level of MyoD is known to be high in the muscle stem cells (or satellite cells), but not in the rest of skeletal muscle tissue. It is not surprising that analyses concerning important regulatory factors are inconclusive. Lastly, the section regarding the hypothetical use of this dataset is highly speculative that is better suited for Discussion and not as a part of the Results section. Together, these concerns greatly reduce the impact of the work.

*Reviewer #2:*

The manuscript by Terry et al. reports the results from transcriptional profiling of 11 different mouse skeletal muscles and comparing these profiles among the mouse muscle groups, rat EDL and soleus and mouse cardiac and smooth muscle. The study clearly shows that there is huge diversity among skeletal muscles with respect to transcriptional profiles. The authors have created a readily accessible database (Muscle DB) to provide other researchers access to these data. This is a carefully performed study that is likely to help determine why select muscles are affected in some muscle diseases. The paper also shows clearly that neither fiber type or developmental history alone can explain the diversity of transcriptional profiles. The data generated in this study are likely to be mined for many years to come. The remarkable diversity in transcriptional profiles also clearly demonstrates that there is no such thing as a truly representative muscles (i.e., EDL as fast twitch and soleus as slow twitch) and hence, when analyzing animal models of humans disease, it will be important to study more muscle groups to truly assess the consequences of disease. The data are also likely to be important for identifying new drug targets. While it would have been even more exciting if a disease model had been included, there is no doubt that this study will contribute greatly to the muscle field.

*Reviewer #3:*

In the current manuscript entitled "Transcriptional profiling reveals extraordinary diversity among skeletal muscle tissues", the authors addressed an important and outstanding issue in the field of skeletal muscle research: how similar or different are the tissues from distinct skeletal muscle groups?

A major strength of this manuscript is that it assessed this issue on a comprehensive level.

Based on their analysis, the authors concluded that distinct skeletal muscle groups are profoundly different in their gene expression profiles, and provided some useful and novel examples to understand these differences and exploit this information for potential advancement of our understanding of muscle disease pathogenesis and improvement of disease treatment. Overall, the current manuscript is of high relevance and importance to the skeletal muscle field. However, as a Resource paper, it may need to develop or implement new technologies/computational models and potential validation experiments to be of high-level interest to the broader audience.

My major concern is whether this manuscript is more appropriately suited in a specialized journal.

1) Due to the tissue harvesting procedures and the nature of bulk RNA-Seq used in this manuscript, it is likely that at least part of the differences observed among distinct skeletal muscle groups could be due to the other cell types present in the muscle tissues rather than from the skeletal muscles per se. Confirmation in MSCs/FAPs and endothelial cells versus purified single fibers from EDL and FDB would provide support.

2) A considerable number of conclusions are solely based on the RNA-Seq data without further experimental validation. For example, in Figure 4 the authors showed two examples of genes that can distinguish DIA and FDB, respectively, from all the other skeletal muscle types examined. The authors should also provide additional evidence to confirm their findings, such as RNA in situ hybridization or tissue IHC. As unique skeletal muscle type markers can be very valuable to the field, I suggest the authors to perform a more systematic analysis to find unique markers for all the skeletal muscle types in their study, if possible, and back up some of the markers using in situ or IHC as suggested above.

3) To obtain a better overview of transcriptional differences among muscle groups, it is advisable to perform GO analysis to examine what functionally-related gene sets are preferentially enriched among distinct skeletal muscle groups.

4) As transcription factors (TFs) commonly serve as nodes of gene expression regulation, it is recommended to perform TF network analysis to examine major TFs/TF networks that drive the transcriptional differences among muscle groups.

5) WGCNA analysis can be performed to examine the major and unique sets of genes that define each muscle groups. It can be helpful to include also smooth and cardiac muscles as an internal control for the analysis.

6) As a Resource paper, the authors should list all the differentially expressed genes and novel splicing events in an excel spreadsheet.

---

## [Author Response]

In the current manuscript entitled "Transcriptional profiling reveals extraordinary diversity among skeletal muscle tissues", the authors addressed an important and outstanding issue in the field of skeletal muscle research: how similar or different are the tissues from distinct skeletal muscle groups?Based on their analysis, the authors concluded that distinct skeletal muscle groups are profoundly different in their gene expression profiles and provided some useful and novel examples to understand these differences. Overall, the current manuscript is of high relevance and importance to the skeletal muscle field. However, as a Resource paper, it may need to develop or implement new technologies/computational models and potential validation experiments to be of high-level interest to the broader audience. The data suggests that neither developmental history nor fiber composition could explain transcriptome diversity among different muscles and the question was left unanswered-based on the dataset it is possible to extract correlates that could explain the diversity.

We agree that the mechanisms governing transcriptional specialization of skeletal muscle are presently unknown and likely to be complex. We argue that a complete understanding of this requires the concerted efforts of many different labs using diverse technical approaches. At a minimum, one would like to see genetic manipulation of key muscle regulatory factors in mice, and subsequent observations of how those changes influence the molecular, physiological, and histological identity of different skeletal muscles. The significance of our Resource is to point the field towards this critical unanswered question and to identify candidate regulatory genes for future studies. Our revised manuscript now contains a variety of pathway analyses, identifying 20 high-confidence candidate transcription factors that may drive muscle specialization (Figure 5 and Table 2).

A major strength of this manuscript is that it assessed this issue on a comprehensive level.A major concern is whether this manuscript is more appropriately suited in a specialized journal.

Regarding concerns that this article belongs in a more specialized journal, we respectfully disagree. Every major gene expression atlas published to date has profiled at most one “representative” skeletal muscle tissue. The reviewers unanimously agree that our data contradicts this implicit assumption. Nevertheless, the myth that skeletal muscle is homogeneous has been perpetuated in the top echelon of academic journals, including Science, Nature, Cell, and PNAS. Such a pervasive blind spot limits both the functional genomics and skeletal muscle fields. Moreover, the data presented herein will find application in the fields of developmental biology and tissue engineering. Challenging a significant and systematic error in the literature is entirely consistent with the publicly stated aims of *eLife*’s editorial leadership.

Major issues:1) Due to the tissue harvesting procedures and the nature of bulk RNA-Seq used in this manuscript, it is likely that at least part of the differences observed among distinct skeletal muscle groups could be due to the other cell types present in the muscle tissues rather than from the skeletal muscles per se. Confirmation in MSCs/FAPs and endothelial cells versus purified single fibers from EDL and FDB would provide support.

We acknowledge that some differential expression in our data is likely due to other cell types and emphasize that to maximize the broad applicability of this Resource, we have chosen to replicate the most common experimental design in the field. Furthermore, we do not believe this caveat calls into question the observation that skeletal muscle transcriptomes are highly divergent, since the sheer magnitude of differential expression in these data is unlikely to come from a minority of nearby cells.

To address the reviewers’ concern directly, we have performed principal component (PC) analysis on the transcriptomes of every tissue studied (Figure 2). We find PC1 accounts for ~80% of variance in the entire dataset, and that PC1 separates different skeletal muscles. In contrast, PC2 accounts for ~8% of variance, and differentiates smooth from cardiac muscles. Naturally, every remaining PC contributed less to the variance than PC1 and PC2. We then measured component loading for a number of RNA markers of different constituent cell types against PC1. We note that *Pecam1*, a marker of endothelial cells, had the greatest correlation with PC1 of all marker genes (R^2^ value of ~0.5). Moreover, we note that the top 100 genes correlated with PC1 (min R^2^ value = 0.84; max R^2^ = 0.99) were almost unanimously canonical skeletal muscle genes (Supplementary file 3). Extending on this observation, we observed that some of these genes were negatively correlated with PC1 (i.e., there were both positive and negative r values), which contradicts the trivial explanation that PC1 is simply a measure of the abundance of muscle cells relative to contaminating cell types in any given sample.

We therefore conclude that the majority of transcriptional differences in this dataset are the result of genuinely different transcriptional programs in skeletal muscle cells. We acknowledge in the Results and Discussion that differential cellular composition, especially from endothelial cells, may contribute to tissue identity; an observation that – by itself – may be of further interest to the field.

2) A considerable number of conclusions are solely based on the RNA-Seq data without further experimental validation. For example, in Figure 4 the authors showed two examples of genes that can distinguish DIA and FDB, respectively, from all the other skeletal muscle types examined. The authors should also provide additional evidence to confirm their findings, such as RNA in situ hybridization or tissue IHC. As unique skeletal muscle type markers can be very valuable to the field, I suggest the authors to perform a more systematic analysis to find unique markers for all the skeletal muscle types in their study, and back up some of the markers using in situ or IHC as suggested above.

Every reasonable step has been taken to cross-validate the accuracy of our RNA-seq data, including qPCR validation of biologically independent samples (Figure 1—figure supplement 4), analysis of replicate reproducibility (Figure 1—figure supplement 3), and computational simulations of appropriate read depth (Figure 1—figure supplement 2). Furthermore, whenever possible, we have tested our data against the literature for both RNA and protein expression of known internal controls (Figures 3, 4, Figure 1—figure supplement 4, and Figure 3—figure supplement 1). We acknowledge that RNA expression does not correlate linearly with protein expression in both the Results and Discussion, but we emphasize that only RNAseq has the dynamic range and throughput to execute a study of this magnitude with any degree of accuracy or reproducibility.

The RNAseq analysis did identify some unique genes expressed in selected skeletal muscles and in response to the Reviewer we used immunohistochemistry to localize expression of Bbox1, enriched in the FDB muscles, and Upk1b, enriched in the diaphragm. We purchased and tested commercially available antibodies to these two proteins but we were not able to obtain images of even positive control tissues such as the bladder for Upk1b.

The primary challenge is that these antibodies are not commonly used, as evidenced by the lack of references in the literature from tissue samples. These genes are not well studied in any tissue and the antibodies have not been validated for IHC on frozen sections. For Upk1b, a Pubmed search pulled up a total of 59 references with the predominant tissue being bladder and 0 references when including skeletal muscle. The antibody data sheet showed staining in human bladder but listed no publications in which the antibody was used. For Bbox1, a Pubmed search yielded 18 total references and none with skeletal muscle. The antibody to Bbox1 is available through Novus and we tested it on many different muscle and non-muscle mouse tissues (frozen sections) and we obtained no specific staining. It should be noted that there are no published papers using this antibody and the western blot data presented comes from cell extracts of unknown origin.

Recognizing this uncertainty, we have moved the tissue specificity figure to the supplements (Figure 1—figure supplement 6), and we acknowledge in the Results that future studies to generate muscle-specific promoter lines will require validation of these observations at the protein level. To assist follow-up studies, we now provide a table listing the most specific gene in every skeletal muscle tissue as the reviewer requested (Supplementary file 6). We further note that some of the most specific genes, such as *Myh7* in the soleus, have been extensively validated using IHC and Western Blot analysis (e.g., Burkholder et al. (1994)).

Finally, we emphasize that although the observation of tissue-specific gene expression supports our overall contention that skeletal muscles have distinct transcriptional programs, validation via IHC is not necessary to confirm this paper’s essential observation regarding specialization of RNA profiles, especially in light of the principal component analysis described above (point #1). Instead, we argue that these tissue-specific expression data should be evaluated as a Resource for generating mouse genetic reagents in follow-up studies, and that the most likely application – generating promoter:Cre fusions – depend more on RNA expression patterns than protein.

3) To obtain a better overview of transcriptional differences among muscle groups, it is advisable to perform GO analysis to examine what functionally-related gene sets are preferentially enriched among distinct skeletal muscle groups.

We used NCBI’s DAVID to identify enriched GO terms in every pairwise comparison (N = 110) of differentially expressed genes in skeletal muscle tissues. We then consolidated these enriched pathways based on the number of times they occur for each tissue. These analyses are now included as Figure 2—figure supplement 1. We found that a variety of terms related to structural components of the sarcomere (Z-disc, A-band, etc.) were common between different skeletal muscle tissues, as well as terms involved in extracellular matrix components, such as Heparin, Collagen, and Integrin. These observations are consistent with the genes most strongly correlated with sample identity being differentially expressed genes of skeletal muscle origin, as discussed in point #1 above.

4) As transcription factors (TFs) commonly serve as nodes of gene expression regulation, it is recommended to perform TF network analysis to examine major TFs/TF networks that drive the transcriptional differences among muscle groups.

We used the same lists of differentially expressed genes from pairwise tissue comparisons (N = 110) as described in Point #3 above in conjunction with Ingenuity Pathway Analysis to predict transcription factors upstream of differentially expressed genes for each skeletal muscle tissue. We then consolidated this list by identifying transcription factors predicted to be upstream of the differentially expressed genes in multiple tissues, as well as being themselves differentially regulated. Table 2 summarizes these data and presents 20 high confidence transcription factors as well as the functionally significant muscle genes with whom they interact.

These data are presented in a visual network diagram in Figure 5. Key differentially expressed genes, such as *Myostatin* and *Vegfa*, are included with the transcription factors predicted to regulate their expression. Additionally, we show links between these genes and known regulatory events influencing muscle disease and physiology.

5) WGCNA analysis can be performed to examine the major and unique sets of genes that define each muscle groups. It can be helpful to include also smooth and cardiac muscles as an internal control for the analysis.

We performed WGCNA analysis using two different sets of input data: (1) expression data condensed into the primary groups of cardiac, smooth, skeletal muscle cluster 1, and skeletal muscle cluster 2, and (2) expression data from all 17 mouse tissues (Figure 2—figure supplement 2). The identities of the most statistically significant clusters of genes are available in Supplementary files 4 and 5. GO analysis of these gene lists revealed that the most statistically significant clusters were enriched in skeletal muscle-specific genes, consistent with points #1 (PCA) and #3 (GO analysis) described above.

6) Try to identify molecular correlates of muscle diversity.

Principal component analysis (point #1 above) revealed that the genes most strongly correlated with muscle identity are conventional skeletal muscle genes, including different versions of *Troponin, Tropomyosin, Titin, Calsequestrin, Myosin heavy chain*, and *Myosin light chain* (Supplementary file 3). Furthermore, we used pathway analyses (point #4 above) to predict the transcription factors upstream of these genes (Table 2). This study revealed an interconnected web of transcription factors, effector genes, and muscle disease (Figure 5). The available data suggest that the relative expression of various characteristic skeletal muscle genes determines tissue specialization, and further suggests candidate transcription factors that may drive the establishment and maintenance of these transcriptional programs. We make special note of *Smarca4*, the most statistically significant candidate TF. *Smarca4* activates muscle gene transcription, suggesting a link between muscle differentiation and specialization in adulthood (Albini et al. (2015)). These new results suggest novel hypotheses that may be tested regarding the molecular drivers of muscle diversity and underscore the importance that this dataset be made available to the community.

7) As a Resource, paper the authors should list all the differentially expressed genes and novel splicing events in an excel spreadsheet.

All differentially expressed transcripts among skeletal muscles are now available as Supplementary file 2. All novel splice junctions in mouse skeletal muscle are now available as Supplementary file 7; all novel splice junctions in rat skeletal muscle are now available as Supplementary file 8. The *.bed files with marked novel junctions are also available in the NCBI GEO submission.